# Correlation Clustering with Adaptive Similarity Queries

**Marco Bressan**
Department of Computer Science
University of Rome Sapienza

**Nicolò Cesa-Bianchi**
Department of Computer Science & DSRC
Università degli Studi di Milano

**Andrea Paudice**
Department of Computer Science
Università degli Studi di Milano & IIT

**Fabio Vitale**
Department of Computer Science
University of Lille & Inria

## Abstract

In correlation clustering, we are given $n$ objects together with a binary similarity score between each pair of them. The goal is to partition the objects into clusters so to minimise the disagreements with the scores. In this work we investigate correlation clustering as an active learning problem: each similarity score can be learned by making a query, and the goal is to minimise both the disagreements and the total number of queries. On the one hand, we describe simple active learning algorithms, which provably achieve an almost optimal trade-off while giving cluster recovery guarantees, and we test them on different datasets. On the other hand, we prove information-theoretical bounds on the number of queries necessary to guarantee a prescribed disagreement bound. These results give a rich characterization of the trade-off between queries and clustering error.

## 1 Introduction

Clustering is a central problem in unsupervised learning. A clustering problem is typically represented by a set of elements together with a notion of similarity (or dissimilarity) between them. When the elements are points in a metric space, dissimilarity can be measured via a distance function. In more general settings, when the elements to be clustered are members of an abstract set $V$, similarity is defined by an arbitrary symmetric function $\sigma$ defined on pairs of distinct elements in $V$. Correlation Clustering (CC) [4] is a well-known special case where $\sigma$ is a $\{-1, +1\}$-valued function establishing whether any two distinct elements of $V$ are similar or not. The objective of CC is to cluster the points in $V$ so to maximize the correlation with $\sigma$. More precisely, CC seeks a clustering minimizing the number of errors, where an error is given by any pair of elements having similarity $-1$ and belonging to the same cluster, or having similarity $+1$ and belonging to different clusters. Importantly, there are no a priori limitations on the number of clusters or their sizes: all partitions of $V$, including the trivial ones, are valid. Given $V$ and $\sigma$, the error achieved by an optimal clustering is known as the *Correlation Clustering index*, denoted by OPT. A convenient way of representing $\sigma$ is through a graph $G = (V, E)$ where $\{u, v\} \in E$ iff $\sigma(u, v) = +1$. Note that OPT $= 0$ is equivalent to a perfectly clusterable graph (i.e., $G$ is the union of disjoint cliques). Since its introduction, CC has attracted a lot of interest in the machine learning community, and has found numerous applications in entity resolution [16], image analysis [18], and social media analysis [25]. Known problems in data integration [14] and biology [5] can be cast into the framework of CC [26].

From a machine learning viewpoint, we are interested in settings when the similarity function $\sigma$ is not available beforehand, and the algorithm must learn $\sigma$ by querying for its value on pairs of objects. This setting is motivated by scenarios in which the similarity information is costly to obtain. For

example, in entity resolution, disambiguating between two entities may require invoking the user's help. Similarly, deciding if two documents are similar may require a complex computation, and possibly the interaction with human experts. In these active learning settings, the learner's goal is to trade the clustering error against the number of queries. Hence, the fundamental question is: how many queries are needed to achieve a specified clustering error? Or, in other terms, how close can we get to OPT, under a prescribed query budget $Q$?

## 1.1 Our Contributions

In this work we characterize the trade-off between the number $Q$ of queries and the clustering error on $n$ points. The table below here summarizes our bounds in the context of previous work. Running time and upper/lower bounds on the expected clustering error are expressed in terms of the number of queries $Q$, and all our upper bounds assume $Q = \Omega(n)$ while our lower bounds assume $Q = \mathcal{O}(n^2)$.

| Running time | Expected clustering error | Reference |
|---|---|---|
| $Q$ + LP solver + rounding | $3(\ln n + 1)\text{OPT} + \mathcal{O}(n^{5/2}/\sqrt{Q})$ | [7] |
| $Q$ | $3\text{OPT} + \mathcal{O}(n^3/Q)$ | Theorem 1 (see also [6]) |
| Exponential | $\text{OPT} + \mathcal{O}(n^{5/2}/\sqrt{Q})$ | Theorem 7 |
| Exponential (OPT $= 0$) | $\widetilde{\mathcal{O}}(n^3/Q)$ | Theorem 7 |
| Unrestricted (OPT $= 0$) | $\Omega(n^2/\sqrt{Q})$ | Theorem 8 |
| Unrestricted (OPT $\gg 0$) | $\text{OPT} + \Omega(n^3/Q)$ | Theorem 9 |

Our first set of contributions is algorithmic. We take inspiration from an existing greedy algorithm, KwikCluster [2], that has expected error $3\text{OPT}$ but a vacuous $\mathcal{O}(n^2)$ worst-case bound on the number of queries. We propose a variant of KwikCluster, called ACC, for which we prove several desirable properties. First, ACC achieves expected clustering error $3\text{OPT} + \mathcal{O}(n^3/Q)$, where $Q = \Omega(n)$ is a deterministic bound on the number of queries. In particular, if ACC is run with $Q = \binom{n}{2}$, then it becomes exactly equivalent to KwikCluster. Second, ACC recovers adversarially perturbed latent clusters. More precisely, if the input contains a cluster $C$ obtained from a clique by adversarially perturbing a fraction $\varepsilon$ of its edges (internal to the clique or leaving the clique), then ACC returns a cluster $\widehat{C}$ such that $\mathbb{E}\big[|C \oplus \widehat{C}|\big] = \mathcal{O}(\varepsilon|C| + n^2/Q)$, where $\oplus$ denotes symmetric difference. This means that ACC recovers almost completely all perturbed clusters that are large enough to be "seen" with $Q$ queries. We also show, under stronger assumptions, that via independent executions of ACC one can recover exactly all large clusters with high probability. Third, we show a variant of ACC, called ACCESS (for Early Stopping Strategy), that makes significantly less queries on some graphs. For example, when OPT $= 0$ and there are $\Omega(n^3/Q)$ similar pairs, the expected number of queries made by ACCESS is only the square root of the queries made by ACC. In exchange, ACCESS makes at most $Q$ queries in expectation rather than deterministically.

Our second set of contributions is a nearly complete information-theoretic characterization of the query vs. clustering error trade-off (thus, ignoring computational efficiency). Using VC theory, we prove that for all $Q = \Omega(n)$ the strategy of minimizing disagreements on a random subset of pairs achieves, with high probability, clustering error bounded by $\text{OPT} + \mathcal{O}(n^{5/2}/\sqrt{Q})$, which reduces to $\widetilde{\mathcal{O}}(n^3/Q)$ when OPT $= 0$. The VC theory approach can be applied to any efficient approximation algorithm, too. The catch is that the approximation algorithm cannot ask the similarity of arbitrary pairs, but only of pairs included in the random sample of edges. The best known approximation factor in this case is $3(\ln n + 1)$ [15], which gives a clustering error bound of $3(\ln n + 1)\text{OPT} + \mathcal{O}(n^{5/2}/\sqrt{Q})$ with high probability. This was already observed in [7] albeit in a slightly different context.

We complement our upper bounds by developing two information-theoretic lower bounds; these lower bounds apply to any algorithm issuing $Q = \mathcal{O}(n^2)$ queries, possibly chosen in an adaptive way. For the general case, we show that any algorithm must suffer an expected clustering error of at least $\text{OPT} + \Omega(n^3/Q)$. In particular, for $Q = \Theta(n^2)$ any algorithm still suffers an additive error of order $n$, and for $Q = \Omega(n)$ our algorithm ACC is essentially optimal in its additive error term. For the special case OPT $= 0$, we show a lower bound $\Omega(n^2/\sqrt{Q})$.

Finally, we evaluate our algorithms empirically on real-world and synthetic datasets.

## 2 Related work

Minimizing the correlation clustering error is APX-hard [9], and the best efficient algorithm found so far achieves $2.06\,\mathrm{OPT}$ [10]. This almost matches the best possible approximation factor 2 achievable via the natural LP relaxation of the problem [9]. A very simple and elegant algorithm for approximating CC is KwikCluster [2]. At each round, KwikCluster draws a random pivot $\pi_r$ from $V$, queries the similarities between $\pi_r$ and every other node in $V$, and creates a cluster $C$ containing $\pi_r$ and all points $u$ such that $\sigma(\pi_r, u) = +1$. The algorithm then recursively invokes itself on $V \setminus C$. On any instance of CC, KwikCluster achieves an expected error bounded by $3\mathrm{OPT}$. However, it is easy to see that KwikCluster makes $\Theta(n^2)$ queries in the worst case (e.g., if $\sigma$ is the constant function $-1$). Our algorithms can be seen as a parsimonious version of KwikCluster whose goal is reducing the number of queries.

The work closest to ours is [6]. Their algorithm runs KwikCluster on a random subset of $1/(2\varepsilon)$ nodes and stores the set $\Pi$ of resulting pivots. Then, each node $v \in V \setminus \Pi$ is assigned to the cluster identified by the pivot $\pi \in \Pi$ with smallest index and such that $\sigma(v, \pi) = +1$. If no such pivot is found, then $v$ becomes a singleton cluster. According to [6, Lemma 4.1], the expected clustering error for this variant is $3\mathrm{OPT} + \mathcal{O}(\varepsilon n^2)$, which can be compared to our bound for ACC by setting $Q = n/\varepsilon$. On the other hand our algorithms are much simpler and significantly easier to analyze. This allows us to prove a set of additional properties, such as cluster recovery and instance-dependent query bounds. It is unclear whether these results are obtainable with the techniques of [6].

Another line of work attempts to circumvent computational hardness by using the more powerful same-cluster queries (SCQ). A same-cluster query tells whether any two given nodes are clustered together according to an optimal clustering or not. In [3] SCQs are used to design a FPTAS for a variant of CC with bounded number of clusters. In [23] SCQs are used to design algorithms for solving CC optimally by giving bounds on $Q$ which depend on OPT. Unlike our setting, both works assume *all* $\binom{n}{2}$ similarities are known in advance. The work [21] considers the case in which there is a latent clustering with $\mathrm{OPT} = 0$. The algorithm can issue SCQs, however the oracle is noisy: each query is answered incorrectly with some probability, and the noise is persistent (repeated queries give the same noisy answer). The above setting is closely related to the stochastic block model (SBM), which is a well-studied model for cluster recovery [1, 19, 22]. However, few works investigate SBMs with pairwise queries [12]. Our setting is strictly harder because our oracle has a budget of OPT adversarially incorrect answers.

A different model is edge classification. Here the algorithm is given a graph $\mathcal{G}$ with hidden binary labels on the edges. The task is to predict the sign of all edges by querying as few labels as possible [7, 11, 13]. As before, the oracle can have a budget OPT of incorrect answers, or a latent clustering with $\mathrm{OPT} = 0$ is assumed and the oracle's answers are affected by persistent noise. Unlike correlation clustering, in edge classification the algorithm is not constrained to predict in agreement with a partition of the nodes. On the other hand, the algorithm cannot query arbitrary pairs of nodes in $V$, but only those that form an edge in $\mathcal{G}$.

**Preliminaries and notation.** We denote by $V \equiv \{1, \dots, n\}$ the set of input nodes, by $\mathcal{E} \equiv \binom{V}{2}$ the set of all pairs $\{u, v\}$ of distincts nodes in $V$, and by $\sigma : \mathcal{E} \to \{-1, +1\}$ the binary similarity function. A clustering $\mathcal{C}$ is a partition of $V$ in disjoint clusters $C_i : i = 1, \dots, k$. Given $\mathcal{C}$ and $\sigma$, the set $\Gamma_{\mathcal{C}}$ of mistaken edges contains all pairs $\{u, v\}$ such that $\sigma(u, v) = -1$ and $u, v$ belong to same cluster of $\mathcal{C}$ and all pairs $\{u, v\}$ such that $\sigma(u, v) = +1$ and $u, v$ belong to different clusters of $\mathcal{C}$. The cost $\Delta_{\mathcal{C}}$ of $\mathcal{C}$ is $|\Gamma_{\mathcal{C}}|$. The correlation clustering index is $\mathrm{OPT} = \min_{\mathcal{C}} \Delta_{\mathcal{C}}$, where the minimum is over all clusterings $\mathcal{C}$. We often view $V, \sigma$ as a graph $G = (V, E)$ where $\{u, v\} \in E$ is an edge if and only if $\sigma(u, v) = +1$. In this case, for any subset $U \subseteq V$ we let $G[U]$ be the subgraph of $G$ induced by $U$, and for any $v \in V$ we let $\mathcal{N}_v$ be the neighbor set of $v$.

A triangle is any unordered triple $T = \{u, v, w\} \subseteq V$. We denote by $e = \{u, w\}$ a generic triangle edge; we write $e \subset T$ and $v \in T \setminus e$. We say $T$ is a *bad triangle* if the labels $\sigma(u, v), \sigma(u, w), \sigma(v, w)$ are $\{+, +, -\}$ (the order is irrelevant). We denote by $\mathcal{T}$ the set of all bad triangles in $V$. It is easy to see that the number of edge-disjoint bad triangles is a lower bound on OPT.

Due to space limitations, here most of our results are stated without proof, or with a concise proof sketch; the full proofs can be found in the supplementary material.

## 3 The ACC algorithm

We introduce our active learning algorithm ACC (Active Correlation Clustering).

---
**Algorithm 1** ACC with query rate $f$

---
**Parameters:** residual node set $V_r$, round index $r$
1: **if** $|V_r| = 0$ **then** RETURN
2: **if** $|V_r| = 1$ **then** output singleton cluster $V_r$ and RETURN
3: **if** $r > \lceil f(|V_1| - 1) \rceil$ **then** RETURN
4: Draw pivot $\pi_r$ u.a.r. from $V_r$
5: $C_r \leftarrow \{\pi_r\}$            ▷ Create new cluster and add the pivot to it
6: Draw a random subset $S_r$ of $\lceil f(|V_r| - 1) \rceil$ nodes from $V_r \setminus \{\pi_r\}$
7: **for** each $u \in S_r$ **do** query $\sigma(\pi_r, u)$
8: **if** $\exists\, u \in S_r$ such that $\sigma(\pi_r, u) = +1$ **then**     ▷ Check if there is at least a positive edge
9:      Query all remaining pairs $(\pi_r, u)$ for $u \in V_r \setminus (\{\pi_r\} \cup S_r)$
10:      $C_r \leftarrow C_r \cup \{u : \sigma(\pi_r, u) = +1\}$      ▷ Populate cluster based on queries
11: Output cluster $C_r$
12: ACC$(V_r \setminus C_r, r + 1)$         ▷ Recursive call on the remaining nodes

---

ACC has the same recursive structure as KwikCluster. First, it starts with the full instance $V_1 = V$. Then, for each round $r = 1, 2, \ldots$ it selects a random pivot $\pi_r \in V_r$, queries the similarities between $\pi_r$ and a subset of $V_r$, removes $\pi_r$ and possibly other points from $V_r$, and proceeds on the remaining residual subset $V_{r+1}$. However, while KwikCluster queries $\sigma(\pi_r, u)$ for *all* $u \in V_r \setminus \{\pi_r\}$, ACC queries only $\lceil f(n_r) \rceil \le n_r$ other nodes $u$ (lines 6–7), where $n_r = |V_r| - 1$. Thus, while KwikCluster always finds all positive labels involving the pivot $\pi_r$, ACC can find them or not, with a probability that depends on $f$. The function $f$ is called *query rate function* and dictates the tradeoff between the clustering cost $\Delta$ and the number of queries $Q$, as we prove below. Now, if any of the aforementioned $\lceil f(n_r) \rceil$ queries returns a positive label (line 8), then all the labels between $\pi_r$ and the remaining $u \in V_r$ are queried and the algorithm operates as KwikCluster until the end of the recursive call; otherwise, the pivot becomes a singleton cluster which is removed from the set of nodes. Another important difference is that ACC deterministically stops after at most $\lceil f(n) \rceil$ recursive calls (line 1), declaring all remaining points as singleton clusters. The intuition is that with good probability the clusters not found within $\lceil f(n) \rceil$ rounds are small enough to be safely disregarded. Since the choice of $f$ is delicate, we avoid trivialities by assuming $f$ is positive and smooth enough. Formally:

**Definition 1.** $f : \mathbb{N} \to \mathbb{R}$ *is a* query rate function *if* $f(1) = 1$, *and* $f(n) \le f(n+1) \le \left(1 + \frac{1}{n}\right)f(n)$ *for all* $n \in \mathbb{N}$. *This implies* $\frac{f(n+k)}{n+k} \le \frac{f(n)}{n}$ *for all* $k \ge 1$.

We can now state formally our bounds for ACC.

**Theorem 1.** *For any query rate function $f$ and any labeling $\sigma$ on $n$ nodes, the expected cost $\mathbb{E}[\Delta_A]$ of the clustering output by* ACC *satisfies*

$$\mathbb{E}[\Delta_A] \le 3\text{OPT} + \frac{2e - 1}{2(e - 1)} \frac{n^2}{f(n)} + \frac{n}{e} .$$

*The number of queries made by* ACC *is deterministically bounded as $Q \le n\lceil f(n) \rceil$. In the special case $f(n) = n$ for all $n \in \mathbb{N}$,* ACC *reduces to KwikCluster and achieves $\mathbb{E}[\Delta_A] \le 3\text{OPT}$ with $Q \le n^2$.*

Note that Theorem 1 gives an upper bound on the error achievable when using $Q$ queries: since $Q = nf(n)$, the expected error is at most $3\text{OPT} + \mathcal{O}(n^3/Q)$. Furthermore, as one expects, if the learner is allowed to ask for all edge signs, then the *exact* bound of KwikCluster is recovered (note that the first formula in Theorem 1 clearly does not take into account the special case when $f(n) = n$, which is considered in the last part of the statement).

**Proof sketch.** Look at a generic round $r$, and consider a pair of points $\{u, w\} \in V_r$. The essence is that ACC can misclassify $\{u, w\}$ in one of two ways. First, if $\sigma(u, w) = -1$, ACC can choose as

pivot $\pi_r$ a node $v$ such that $\sigma(v, u) = \sigma(v, w) = +1$. In this case, if the condition on line 8 holds, then ACC will cluster $v$ together with $u$ and $w$, thus mistaking $\{u, w\}$. If instead $\sigma(u, w) = +1$, then ACC could mistake $\{u, w\}$ by pivoting on a node $v$ such that $\sigma(v, u) = +1$ and $\sigma(v, w) = -1$, and clustering together only $v$ and $u$. Crucially, both cases imply the existence of a bad triangle $T = \{u, w, v\}$. We charge each such mistake to exactly one bad triangle $T$, so that no triangle is charged twice. The expected number of mistakes can then be bound by 3OPT using the packing argument of [2] for KwikCluster. Second, if $\sigma(u, w) = +1$ then ACC could choose one of them, say $u$, as pivot $\pi_r$, and assign it to a singleton cluster. This means the condition on line 8 fails. We can then bound the number of such mistakes as follows. Suppose $\pi_r$ has $cn/f(n)$ positive labels towards $V_r$ for some $c \geq 0$. Loosely speaking, we show that the check of line 8 fails with probability $e^{-c}$, in which case $cn/f(n)$ mistakes are added. In expectation, this gives $cne^{-c}/f(n) = \mathcal{O}(n/f(n))$ mistakes. Over all $f(n) \leq n$ rounds, this gives an overall $\mathcal{O}(n^2/f(n))$. (The actual proof has to take into account that all the quantities involved here are not constants, but random variables).

## 3.1    ACC **with Early Stopping Strategy**

We can refine our algorithm ACC so that, in some cases, it takes advantage of the structure of the input to reduce significantly the expected number of queries. To this end we see the input as a graph $G$ with edges corresponding to positive labels (see above). Suppose then $G$ contains a sufficiently small number $\mathcal{O}(n^2/f(n))$ of edges. Since ACC performs up to $\lceil f(n) \rceil$ rounds, it could make $Q = \Theta(f(n)^2)$ queries. However, with just $\lceil f(n) \rceil$ queries one could *detect* that $G$ contains $\mathcal{O}(n^2/f(n))$ edges, and immediately return the trivial clustering formed by all singletons. The expected error would obviously be at most OPT $+ \mathcal{O}(n^2/f(n))$, i.e. the same of Theorem 1. More generally, at each round $r$ with $\lceil f(n_r) \rceil$ queries one can check if the residual graph contains at least $n^2/f(n)$ edges; if the test fails, declaring all nodes in $V_r$ as singletons gives expected additional error $\mathcal{O}(n^2/f(n))$. The resulting algorithm is a variant of ACC that we call ACCESS (ACC with Early Stopping Strategy). The pseudocode can be found in the supplementary material.

First, we show ACCESS gives guarantees virtually identical to ACC (only, with $Q$ in expectation). Formally:

**Theorem 2.** *For any query rate function $f$ and any labeling $\sigma$ on $n$ nodes, the expected cost $\mathbb{E}[\Delta_A]$ of the clustering output by* ACCESS *satisfies*

$$\mathbb{E}[\Delta_A] \leq 3\text{OPT} + 2\frac{n^2}{f(n)} + \frac{n}{e} \ .$$

*Moreover, the expected number of queries performed by* ACCESS *is $\mathbb{E}[Q] \leq n(\lceil f(n) \rceil + 4)$.*

Theorem 2 reassures us that ACCESS is no worse than ACC. In fact, if most edges of $G$ belong to relatively large clusters (namely, all but $O(n^2/f(n))$ edges), then we can show ACCESS uses much fewer queries than ACC (in a nutshell, ACCESS quickly finds all large clusters and then quits). The following theorem captures the essence. For simplicity we assume OPT $= 0$, i.e. $G$ is a disjoint union of cliques.

**Theorem 3.** *Suppose* OPT $= 0$ *so $G$ is a union of disjoint cliques. Let $C_1, \ldots, C_\ell$ be the cliques of $G$ in nondecreasing order of size. Let $i'$ be the smallest $i$ such that $\sum_{j=1}^{i} |E_{C_j}| = \Omega(n^2/f(n))$, and let $h(n) = |C_{i'}|$. Then* ACCESS *makes in expectation $\mathbb{E}[Q] = \mathcal{O}(n^2 \lg(n)/h(n))$ queries.*

As an example, say $f(n) = \sqrt{n}$ and $G$ contains $n^{1/3}$ cliques of $n^{2/3}$ nodes each. Then for ACC Theorem 1 gives $Q \leq nf(n) = \mathcal{O}(n^{3/2})$, while for ACCESS Theorem 3 gives $\mathbb{E}[Q] = \mathcal{O}(n^{4/3} \lg(n))$.

## 4    Cluster recovery

In the previous section we gave bounds on $\mathbb{E}[\Delta]$, the expected *total* cost of the clustering. However, in applications such as community detection and alike, the primary objective is recovering accurately the latent clusters of the graph, the sets of nodes that are "close" to cliques. This is usually referred to as *cluster recovery*. For this problem, an algorithm that outputs a good approximation $\widehat{C}$ of every latent cluster $C$ is preferable to an algorithm that minimizes $\mathbb{E}[\Delta]$ globally. In this section we show that ACC natively outputs clusters that are close to the latent clusters in the graph, thus acting as a

cluster recovery tool. We also show that, for a certain type of latent clusters, one can amplify the accuracy of ACC via independent executions and recover all clusters exactly with high probability.

To capture the notion of "latent cluster", we introduce the concept of $(1-\varepsilon)$-*knit* set. As usual, we view $V, \sigma$ as a graph $G = (V, E)$ with $e \in E$ iff $\sigma(e) = +1$. Let $E_C$ be the edges in the subgraph induced by $C \subseteq V$ and $\mathrm{cut}(C, \overline{C})$ be the edges between $C$ and $\overline{C} = V \setminus C$.

**Definition 2.** *A subset* $C \subseteq V$ *is* $(1-\varepsilon)$-knit *if* $\big|E_C\big| \geq (1-\varepsilon)\binom{|C|}{2}$ *and* $\big|\mathrm{cut}(C, \overline{C})\big| \leq \varepsilon\binom{|C|}{2}$.

Suppose now we have a cluster $\widehat{C}$ as "estimate" of $C$. We quantify the distance between $C$ and $\widehat{C}$ as the cardinality of their symmetric difference, $\big|\widehat{C} \oplus C\big| = \big|\widehat{C} \setminus C\big| + \big|C \setminus \widehat{C}\big|$. The goal is to obtain, for each $(1-\varepsilon)$-knit set $C$ in the graph, a cluster $\widehat{C}$ with $\big|\widehat{C} \oplus C\big| = \mathcal{O}(\varepsilon|C|)$ for some small $\varepsilon$. We prove ACC does exactly this. Clearly, we must accept that if $C$ is too small, i.e. $|C| = o(n/f(n))$, then ACC will miss $C$ entirely. But, for $|C| = \Omega(n/f(n))$, we can prove $\mathbb{E}[|\widehat{C} \oplus C|] = \mathcal{O}(\varepsilon|C|)$. We point out that the property of being $(1-\varepsilon)$-knit is rather weak for an algorithm, like ACC, that is completely oblivious to the global topology of the cluster — all what ACC tries to do is to blindly cluster together all the neighbors of the current pivot. In fact, consider a set $C$ formed by two disjoint cliques of equal size. This set would be close to $1/2$-knit, and yet ACC would never produce a single cluster $\widehat{C}$ corresponding to $C$. Things can only worsen if we consider also the edges in $\mathrm{cut}(C, \overline{C})$, which can lead ACC to assign the nodes of $C$ to several different clusters when pivoting on $\overline{C}$. Hence it is not obvious that a $(1-\varepsilon)$-knit set $C$ can be efficiently recovered by ACC.

Note that this task can be seen as an *adversarial* cluster recovery problem. Initially, we start with a disjoint union of cliques, so that $\mathrm{OPT} = 0$. Then, an adversary flips the signs of some of the edges of the graph. The goal is to retrieve every original clique that has not been perturbed excessively. Note that we put no restriction on how the adversary can flip edges; therefore, this adversarial setting subsumes constrained adversaries. For example, it subsumes the high-probability regime of the stochastic block model [17] where edges are flipped according to some distribution.

We can now state our main cluster recovery bound for ACC.

**Theorem 4.** *For every* $C \subseteq V$ *that is* $(1-\varepsilon)$-*knit,* ACC *outputs a cluster* $\widehat{C}$ *such that* $\mathbb{E}\big[|C \oplus \widehat{C}|\big] \leq 3\varepsilon|C| + \min\big\{\frac{2n}{f(n)}, \big(1 - \frac{f(n)}{n}\big)|C|\big\} + |C|e^{-|C|f(n)/5n}$.

The $\min$ in the bound captures two different regimes: when $f(n)$ is very close to $n$, then $\mathbb{E}\big[|C \oplus \widehat{C}|\big] = \mathcal{O}(\varepsilon|C|)$ independently of the size of $C$, but when $f(n) \ll n$ we need $|C| = \Omega(n/f(n))$, i.e., $|C|$ must be large enough to be found by ACC.

## 4.1  Exact cluster recovery via amplification

For certain latent clusters, one can get recovery guarantees significantly stronger than the ones given natively by ACC (see Theorem 4). We start by introducing *strongly* $(1-\varepsilon)$-*knit* sets (also known as quasi-cliques). Recall that $\mathcal{N}_v$ is the neighbor set of $v$ in the graph $G$ induced by the positive labels.

**Definition 3.** *A subset* $C \subseteq V$ *is* strongly $(1-\varepsilon)$-knit *if, for every* $v \in C$, *we have* $\mathcal{N}_v \subseteq C$ *and* $|\mathcal{N}_v| \geq (1-\varepsilon)(|C| - 1)$.

We remark that ACC alone does not give better guarantees on strongly $(1-\varepsilon)$-knit subsets than on $(1-\varepsilon)$-knit subsets. Suppose for example that $|\mathcal{N}_v| = (1-\varepsilon)(|C| - 1)$ for all $v \in C$. Then $C$ is strongly $(1-\varepsilon)$-knit, and yet when pivoting on any $v \in C$ ACC will inevitably produce a cluster $\widehat{C}$ with $|\widehat{C} \oplus C| \geq \varepsilon|C|$, since the pivot has edges to less than $(1-\varepsilon)|C|$ other nodes of $C$.

To bypass this limitation, we run ACC several times to amplify the probability that every node in $C$ is found. Recall that $V = [n]$. Then, we define the id of a cluster $\widehat{C}$ as the smallest node of $\widehat{C}$. The min-tagging rule is the following: when forming $\widehat{C}$, use its id to tag all of its nodes. Therefore, if $u_{\widehat{C}} = \min\{u \in \widehat{C}\}$ is the id of $\widehat{C}$, we will set $\mathrm{id}(v) = u_{\widehat{C}}$ for every $v \in \widehat{C}$. Consider now the following algorithm, called ACR (Amplified Cluster Recovery). First, ACR performs $K$ independent runs of ACC on input $V$, using the min-tagging rule on each run. In this way, for each $v \in V$ we obtain $K$ tags $\mathrm{id}_1(v), \ldots, \mathrm{id}_K(v)$, one for each run. Thereafter, for each $v \in V$ we select the tag that $v$ has received most often, breaking ties arbitrarily. Finally, nodes with the same tag are clustered

together. One can prove that, with high probability, this clustering contains all strongly $(1 - \varepsilon)$-knit sets. In other words, ACR with high probability recovers all such latent clusters *exactly*. Formally, we prove:

**Theorem 5.** *Let $\varepsilon \leq \frac{1}{10}$ and fix $p > 0$. If ACR is run with $K = 48 \ln \frac{n}{p}$, then the following holds with probability at least $1 - p$: for every strongly $(1 - \varepsilon)$-knit $C$ with $|C| > 10 \frac{n}{f(n)}$, the algorithm outputs a cluster $\widehat{C}$ such that $\widehat{C} = C$.*

It is not immediately clear that one can extend this result by relaxing the notion of strongly $(1-\varepsilon)$-knit set so to allow for edges between $C$ and the rest of the graph. We just notice that, in that case, every node $v \in C$ could have a neighbor $x_v \in V \setminus C$ that is smaller than every node of $C$. In this case, when pivoting on $v$ ACC would tag $v$ with $x$ rather than with $u_C$, disrupting ACR.

# 5    A fully additive scheme

In this section, we introduce a(n inefficient) fully additive approximation algorithm achieving cost $\text{OPT} + n^2 \varepsilon$ in high probability using order of $\frac{n}{\varepsilon^2}$ queries. When $\text{OPT} = 0$, $Q = \frac{n}{\varepsilon} \ln \frac{1}{\varepsilon}$ suffices. Our algorithm combines uniform sampling with empirical risk minimization and is analyzed using VC theory.

First, note that CC can be formulated as an agnostic binary classification problem with binary classifiers $h_{\mathcal{C}} : \mathcal{E} \to \{-1, +1\}$ associated with each clustering $\mathcal{C}$ of $V$ (recall that $\mathcal{E}$ denotes the set of all pairs $\{u, v\}$ of distinct elements $u, v \in V$), and we assume $h_{\mathcal{C}}(u, v) = +1$ iff $u$ and $v$ belong to the same cluster of $\mathcal{C}$. Let $\mathcal{H}_n$ be the set of all such $h_{\mathcal{C}}$. The risk of a classifier $h_{\mathcal{C}}$ with respect to the uniform distribution over $\mathcal{E}$ is $\mathbb{P}(h_{\mathcal{C}}(e) \neq \sigma(e))$ where $e$ is drawn u.a.r. from $\mathcal{E}$. It is easy to see that the risk of any classifier $h_{\mathcal{C}}$ is directly related to $\Delta_{\mathcal{C}}$, $\mathbb{P}(h_{\mathcal{C}}(e) \neq \sigma(e)) = \Delta_{\mathcal{C}} / \binom{n}{2}$. Hence, in particular, $\text{OPT} = \binom{n}{2} \min_{h \in \mathcal{H}_n} \mathbb{P}(h(e) \neq \sigma(e))$. Now, it is well known —see, e.g., [24, Theorem 6.8]— that we can minimize the risk to within an additive term of $\varepsilon$ using the following procedure: query $\mathcal{O}(d/\varepsilon^2)$ edges drawn u.a.r. from $\mathcal{E}$, where $d$ is the VC dimension of $\mathcal{H}_n$, and find the clustering $\mathcal{C}$ such that $h_{\mathcal{C}}$ makes the fewest mistakes on the sample. If there is $h^* \in \mathcal{H}_n$ with zero risk, then $\mathcal{O}((d/\varepsilon) \ln(1/\varepsilon))$ random queries suffice. A trivial upper bound on the VC dimension of $\mathcal{H}_n$ is $\log_2 |\mathcal{H}_n| = \mathcal{O}(n \ln n)$. The next result gives the exact value.

**Theorem 6.** *The VC dimension of the class $\mathcal{H}_n$ of all partitions of $n$ elements is $n - 1$.*

*Proof.* Let $d$ be the VC dimension of $\mathcal{H}_n$. We view an instance of CC as the complete graph $K_n$ with edges labelled by $\sigma$. Let $T$ be any spanning tree of $K_n$. For any labeling $\sigma$, we can find a clustering $\mathcal{C}$ of $V$ such that $h_{\mathcal{C}}$ perfectly classifies the edges of $T$: simply remove the edges with label $-1$ in $T$ and consider the clusters formed by the resulting connected components. Hence $d \geq n - 1$ because any spanning tree has exactly $n - 1$ edges. On the other hand, any set of $n$ edges must contain at least a cycle. It is easy to see that no clustering $\mathcal{C}$ makes $h_{\mathcal{C}}$ consistent with the labeling $\sigma$ that gives positive labels to all edges in the cycle but one. Hence $d < n$. □

An immediate consequence of the above is the following.

**Theorem 7.** *There exists a randomized algorithm A that, for all $0 < \varepsilon < 1$, finds a clustering $\mathcal{C}$ satisfying $\Delta_{\mathcal{C}} \leq \text{OPT} + \mathcal{O}(n^2 \varepsilon)$ with high probability while using $Q = \mathcal{O}\left(\frac{n}{\varepsilon^2}\right)$ queries. Moreover, if $\text{OPT} = 0$, then $Q = \mathcal{O}\left(\frac{n}{\varepsilon} \ln \frac{1}{\varepsilon}\right)$ queries are enough to find a clustering $\mathcal{C}$ satisfying $\Delta_{\mathcal{C}} = \mathcal{O}(n^2 \varepsilon)$.*

# 6    Lower bounds

In this section we give two lower bounds on the expected clustering error of any (possibly randomized) algorithm. The first bound holds for $\text{OPT} = 0$, and applies to algorithms using a deterministically bounded number of queries. This bound is based on a construction from [8, Lemma 11] and related to kernel-based learning.

**Theorem 8.** *For any $\varepsilon > 0$ such that $\frac{1}{\varepsilon}$ is an even integer, and for every (possibly randomized) learning algorithm asking fewer than $\frac{1}{50\varepsilon^2}$ queries with probability 1, there exists a labeling $\sigma$ on $n \geq \frac{16}{\varepsilon} \ln \frac{1}{\varepsilon}$ nodes such that $\text{OPT} = 0$ and the expected cost of the algorithm is at least $\frac{n^2 \varepsilon}{8}$.*

Our second bound relaxed the assumption on OPT. It uses essentially the same construction of [6, Lemma 6.1], giving asymptotically the same guarantees. However, the bound of [6] applies only to a very restricted class of algorithms: namely, those where the number $q_v$ of queries involving any specific node $v \in V$ is deterministically bounded. This rules out a vast class of algorithms, including KwikCluster, ACC, and ACCESS, where the number of queries involving a node is a function of the random choices of the algorithm. Our lower bound is instead fully general: it holds unconditionally for *any* randomized algorithm, with no restriction on what or how many pairs of points are queried.

**Theorem 9.** *Choose any function $\varepsilon = \varepsilon(n)$ such that $\Omega\left(\frac{1}{n}\right) \leq \varepsilon \leq \frac{1}{2}$ and $\frac{1}{\varepsilon} \in \mathbb{N}$. For every (possibly randomized) learning algorithm and any $n_0 > 0$ there exists a labeling $\sigma$ on $n \geq n_0$ nodes such that the algorithm has expected error $\mathbb{E}[\Delta] \geq \mathrm{OPT} + \frac{n^2 \varepsilon}{80}$ whenever its expected number of queries satisfies $\mathbb{E}[Q] < \frac{n}{80\,\varepsilon}$.*

In fact, the bound of Theorem 9 can be put in a more general form: for any constant $c \geq 1$, the expected error is at least $c \cdot \mathrm{OPT} + A(c)$ where $A(c) = \Omega(n^2 \varepsilon)$ is an additive term with constant factors depending on $c$ (see the proof). Thus, our algorithms ACC and ACCESS are essentially optimal in the sense that, for $c = 3$, they guarantee an optimal additive error up to constant factors.

## 7 Experiments

We verify experimentally the tradeoff between clustering cost and number of queries of ACC, using six datasets from [21, 20]. Four datasets come from real-world data, and two are synthetic; all of them provide a ground-truth partitioning of some set $V$ of nodes. Here we show results for one real-world dataset (cora, with $|V|$=1879 and 191 clusters) and one synthetic dataset (skew, with $|V|$=900 and 30 clusters). Results for the remaining datasets are similar and can be found in the supplementary material. Since the original datasets have OPT = 0, we derived perturbed versions where OPT > 0 as follows. First, for each $\eta \in \{0, 0.1, 0.5, 1\}$ we let $p = \eta|E|/\binom{n}{2}$ where $|E|$ is the number of edges (positive labels) in the dataset (so $\eta$ is the expected number of flipped edges measured as a multiple of $|E|$). Then, we flipped the label of each pair of nodes independently with probability $p$. Obviously for $p = 0$ we have the original dataset.

For every dataset and its perturbed versions we then proceeded as follows. For $\alpha = 0, 0.05, ..., 0.95, 1$, we set the query rate function to $f(x) = x^\alpha$. Then we ran 20 independent executions of ACC, and computed the average number of queries $\mu_Q$ and average clustering cost $\mu_\Delta$. The variance was often negligible, but is reported in the full plots in the supplementary material. The tradeoff between $\mu_\Delta$ and $\mu_Q$ is depicted in Figure 1, where the circular marker highlights the case $f(x) = x$, i.e. KwikCluster.

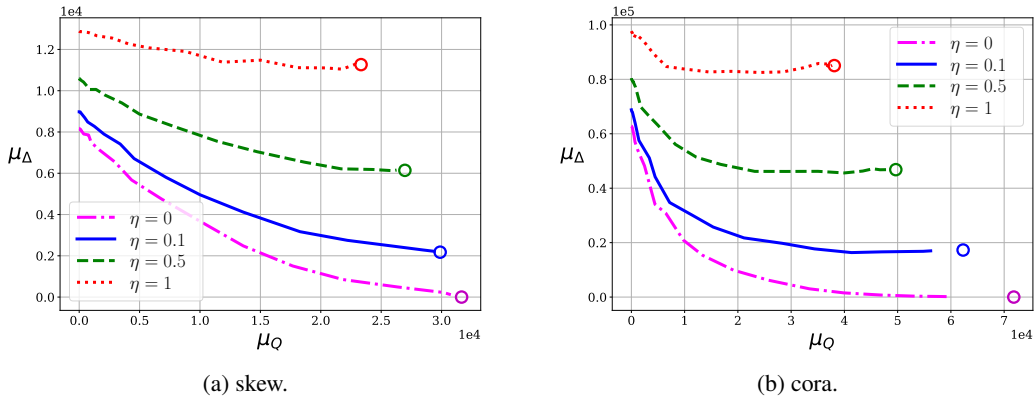

(a) skew.  (b) cora.

Figure 1: Performance of ACC.

The clustering cost clearly drops as the number of queries increases. This drop is particularly marked on cora, where ACC achieves a clustering cost close to that of KwikCluster using an order of magnitude fewer queries. It is also worth noting that, for the case OPT = 0, the measured clustering cost achieved by ACC is 2 to 3 times lower than the theoretical bound of $\approx 3.8n^3/Q$ given by Theorem 1.

**Acknowledgements**

The authors gratefully acknowledge partial support by the Google Focused Award "Algorithms and Learning for AI" (ALL4AI). Marco Bressan and Fabio Vitale are also supported in part by the ERC Starting Grant DMAP 680153 and by the "Dipartimenti di Eccellenza 2018-2022" grant awarded to the Department of Computer Science of the Sapienza University of Rome. Nicolò Cesa-Bianchi is also supported by the MIUR PRIN grant *Algorithms, Games, and Digital Markets* (ALGADIMAR).

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
