[Supplementary Material]

# Correlation Clustering
# with Adaptive Similarity Queries
# (Supplementary Material)

## 1 Probability bounds

We give Chernoff-type probability bounds that can be found in e.g. [2] and that we repeatedly use in our proofs. Let $X_1, \ldots, X_n$ be binary random variables. We say that $X_1, \ldots, X_n$ are non-positively correlated if for all $I \subseteq \{1, \ldots, n\}$ we have:

$$\mathbb{P}[\forall i \in I : X_i = 0] \leq \prod_{i \in I} \mathbb{P}[X_i = 0] \quad \text{and} \quad \mathbb{P}[\forall i \in I : X_i = 1] \leq \prod_{i \in I} \mathbb{P}[X_i = 1] \tag{1}$$

The following holds:

**Lemma 1.** *Let $X_1, \ldots, X_n$ be independent or, more generally, non-positively correlated binary random variables. Let $a_1, \ldots, a_n \in [0, 1]$ and $X = \sum_{i=1}^{n} a_i X_i$. Then, for any $\delta > 0$, we have:*

$$\mathbb{P}[X < (1 - \delta)\mathbb{E}[X]] < e^{-\frac{\delta^2}{2}\mathbb{E}[X]} \tag{2}$$

$$\mathbb{P}[X > (1 + \delta)\mathbb{E}[X]] < e^{-\frac{\delta^2}{2+\delta}\mathbb{E}[X]} \tag{3}$$

## 2 Supplementary Material for Section 3

### 2.1 Pseudocode of ACC

For ease of reference we report the pseudocode of ACC below.

---
**Algorithm 1** ACC with query rate $f$
---
**Parameters:** residual node set $V_r$, round index $r$
 1: **if** $|V_r| = 0$ **then** RETURN
 2: **if** $|V_r| = 1$ **then** output singleton cluster $V_r$ and RETURN
 3: **if** $r > \lceil f(|V_1| - 1) \rceil$ **then** RETURN
 4: Draw pivot $\pi_r$ u.a.r. from $V_r$
 5: $C_r \leftarrow \{\pi_r\}$            ▷ Create new cluster and add the pivot to it
 6: Draw a random subset $S_r$ of $\lceil f(|V_r| - 1) \rceil$ nodes from $V_r \setminus \{\pi_r\}$
 7: **for** each $u \in S_r$ **do** query $\sigma(\pi_r, u)$
 8: **if** $\exists\, u \in S_r$ such that $\sigma(\pi_r, u) = +1$ **then**       ▷ Check if there is at least a positive edge
 9:     Query all remaining pairs $(\pi_r, u)$ for $u \in V_r \setminus (\{\pi_r\} \cup S_r)$
10:     $C_r \leftarrow C_r \cup \{u : \sigma(\pi_r, u) = +1\}$        ▷ Populate cluster based on queries
11: Output cluster $C_r$
12: ACC$(V_r \setminus C_r, r + 1)$           ▷ Recursive call on the remaining nodes
---

## 2.2 Proof of Theorem 1

We refer to the pseudocode above (Algorithm 1). We use $V_r$ to denote the set of remaining nodes at the beginning of the $r$-th recursive call, and we let $n_r = |V_r| - 1$. Hence $V_1 = V$ and $n_1 = n - 1$. If the condition in the **if** statement on line 8 is not true, then $C_r$ is a singleton cluster. We denote by $V_{\text{sing}}$ the set nodes that are output as singleton clusters.

Let $\Gamma_A$ be the set of mistaken edges for the clustering output by ACC and let $\Delta_A = |\Gamma_A|$ be the cost of this clustering. Note that, in any recursive call, ACC misclassifies an edge $e = \{u, w\}$ if and only if $e$ is part of a bad triangle whose third node $v$ is chosen as pivot and does not become a singleton cluster, or if $\sigma(e) = +1$ and at least one of $u, w$ becomes a singleton cluster. More formally, ACC misclassifies an edge $e = \{u, w\}$ if and only if one of the following three disjoint events holds:

$B_1(e)$: There exists $r \leq \lceil f(n-1) \rceil$ and a bad triangle $T \equiv \{u, v, w\} \subseteq V_r$ such that $\pi_r = v$ and $v \notin V_{\text{sing}}$.

$B_2(e)$: There exists $r \leq \lceil f(n-1) \rceil$ such that $u, w \in V_r$ with $\sigma(u, w) = +1$ and $\pi_r \in \{u, w\} \cap V_{\text{sing}}$.

$B_3(e)$: ACC stops after $\lceil f(n-1) \rceil$ rounds without removing neither $u$ nor $w$, and $\sigma(u, w) = +1$.

Therefore the indicator variable for the event "$e$ is mistaken" is:

$$\mathbb{I}\{e \in \Gamma_A\} = \mathbb{I}\{B_1(e)\} + \mathbb{I}\{B_2(e)\} + \mathbb{I}\{B_3(e)\}$$

The expected cost of the clustering is therefore:

$$\mathbb{E}[\Delta_A] = \sum_{e \in \mathcal{E}} \mathbb{P}(B_1(e)) + \sum_{e \in \mathcal{E}} \mathbb{P}(B_2(e)) + \sum_{e \in \mathcal{E}} \mathbb{P}(B_3(e)) \tag{4}$$

We proceed to bound the three terms separately.

**Bounding $\sum_{e \in \mathcal{E}} \mathbb{P}(B_1(e))$.** Fix an arbitrary edge $e = \{u, w\}$. Note that, if $B_1(e)$ occurs, then $T$ is unique, i.e. exactly one bad triangle $T$ in $V$ satisfies the definition of $B_1(e)$. Each occurrence of $B_1(e)$ can thus be charged to a single bad triangle $T$. We may thus write

$$\sum_{e \in \mathcal{E}} \mathbb{I}\{B_1(e)\} = \sum_{e \in \mathcal{E}} \mathbb{I}\{(\exists r)(\exists T \in \mathcal{T}) : T \subseteq V_r \wedge e \subset T \wedge \pi_r \in T \setminus e \wedge \pi_r \notin V_{\text{sing}}\}$$

$$= \sum_{T \in \mathcal{T}} \mathbb{I}\{(\exists r) : T \subseteq V_r \wedge \pi_r \in T \wedge \pi_r \notin V_{\text{sing}}\}$$

$$\leq \sum_{T \in \mathcal{T}} \mathbb{I}\{A_T\}$$

where $A_T \equiv \{(\exists r) : T \subseteq V_r \wedge \pi_r \in T\}$. Let us then bound $\sum_{T \in \mathcal{T}} \mathbb{P}(A_T)$. Let $\mathcal{T}(e) \equiv \{T' \in \mathcal{T} : e \in T'\}$. We use the following fact extracted from the proof of [1, Theorem 6.1]. If $\{\beta_T \geq 0 : T \in \mathcal{T}\}$ is a set of weights on the bad triangles such that $\sum_{T \in \mathcal{T}(e)} \beta_T \leq 1$ for all $e \in \mathcal{E}$, then $\sum_{T \in \mathcal{T}} \beta_T \leq \text{OPT}$. Given $e \in \mathcal{E}$ and $T \in \mathcal{T}$, let $F_T(e)$ be the event corresponding to $T$ being the first triangle in the set $\mathcal{T}(e)$ such that $T \in V_r$ and $\pi_r \in T \setminus e$ for some $r$. Now if $F_T(e)$ holds then $A_T$ holds and no other $A_{T'}$ for $T' \in \mathcal{T}(e) \setminus \{T\}$ holds. Therefore

$$\sum_{T \in \mathcal{T}(e)} \mathbb{I}\{A_T \wedge F_T(e)\} = 1 .$$

If $A_T$ holds for some $r_0$, then it cannot hold for any other $r > r_0$ because $\pi_{r_0} \in T$ implies that for all $r > r_0$ we have $\pi_{r_0} \notin V_r$ implying $T \not\subseteq V_r$. Hence, given that $A_T$ holds for $r_0$, if $F_T(e)$ holds too, then it holds for the same $r_0$ by construction. This implies that $\mathbb{P}(F_T(e) \mid A_T) = \frac{1}{3}$ because ACC chooses the pivot u.a.r. from the nodes in $V_{r_0}$. Thus, for each $e \in E$ we can write

$$1 = \sum_{T \in \mathcal{T}(e)} \mathbb{P}(A_T \wedge F_T(e)) = \sum_{T \in \mathcal{T}(e)} \mathbb{P}(F_T(e) \mid A_T)\mathbb{P}(A_T) = \sum_{T \in \mathcal{T}(e)} \frac{1}{3}\mathbb{P}(A_T) . \tag{5}$$

Choosing $\beta_T = \frac{1}{3}\mathbb{P}(A_T)$ we get $\sum_{T \in \mathcal{T}} \mathbb{P}(A_T) \leq 3\text{OPT}$.

In the proof of KwikCluster, the condition $\sum_{T \in \mathcal{T}(e)} \beta_T \leq 1$ was ensured by considering events $G_T(e) = A_T \wedge e \in \Gamma_A$. Indeed, in KwikCluster the events $\{G_T(e) : T \in \mathcal{T}(e)\}$ are disjoint,

because $G_T(e)$ holds iff $T$ is the first and only triangle in $\mathcal{T}(e)$ whose node opposite to $e$ is chosen as pivot. For ACC this is not true because a pivot can become a singleton cluster, which does not cause $e \in \Gamma_A$ necessarily to hold.

**Bounding $\sum_{e \in \mathcal{E}} \mathbb{P}(B_2(e))$.** For any $u \in V_r$, let $d_r^+(u) = \big| \{v \in V_r \,:\, \sigma(u,v) = +1\} \big|$. We have:

$$\sum_{e \in \mathcal{E}} \mathbb{I}\{B_2(e)\} = \frac{1}{2} \sum_{u \in V} \sum_{r=1}^{\lceil f(n-1) \rceil} \mathbb{I}\{\pi_r = u \wedge \pi_r \in V_{\text{sing}}\} \, d_r^+(u) \,.$$

Taking expectations with respect to the randomization of ACC,

$$\sum_{e \in \mathcal{E}} \mathbb{P}(B_2(e)) = \frac{1}{2} \sum_{u \in V} \sum_{r=1}^{\lceil f(n-1) \rceil} \mathbb{E}\Big[\mathbb{I}\{\pi_r = u \wedge \pi_r \in V_{\text{sing}}\} \, d_r^+(u)\Big]$$

$$= \frac{1}{2} \sum_{u \in V} \sum_{r=1}^{\lceil f(n-1) \rceil} \mathbb{E}\Big[\mathbb{I}\{\pi_r \in V_{\text{sing}}\} \, d_r^+(u) \,\Big|\, \pi_r = u\Big] \mathbb{P}(\pi_r = u)$$

For any round $r$, let $H_{r-1}$ be the sequence of random draws made by the algorithm before round $r$. Then $\mathbb{P}(\pi_r \in V_{\text{sing}} \,|\, \pi_r = u, H_{r-1}) d_r^+(u) = 0$ if either $d_r^+(u) = 0$, or $d_r^+(u) \geq 1$ and $d_r^-(u) < \lceil f(n_r) \rceil$. Otherwise,

$$\mathbb{P}(\pi_r \in V_{\text{sing}} \,|\, \pi_r = u, H_{r-1}) = \prod_{j=0}^{\lceil f(n_r) \rceil - 1} \frac{d_r^-(u) - j}{n_r - j} \leq \left(\frac{d_r^-(u)}{n_r}\right)^{\lceil f(n_r) \rceil} = \left(1 - \frac{d_r^+(u)}{n_r}\right)^{\lceil f(n_r) \rceil}$$

$$(6)$$

where the inequality holds because $d_r^-(u) \leq n_r$. Therefore, when $d_r^+(u) \geq 1$ and $d_r^-(u) \geq \lceil f(n_r) \rceil$,

$$\mathbb{E}\Big[\mathbb{I}\{\pi_r \in V_{\text{sing}}\} \, d_r^+(u) \,\Big|\, \pi_r = u, H_{r-1}\Big] = \mathbb{P}(\pi_r \in V_{\text{sing}} \,|\, \pi_r = u, H_{r-1}) d_r^+(u)$$

$$= \left(1 - \frac{d_r^+(u)}{n_r}\right)^{\lceil f(n_r) \rceil} d_r^+(u)$$

$$= \left(1 - \frac{d_r^+(u)}{n_r}\right)^{\lceil f(n_r) \rceil} d_r^+(u)$$

$$\leq \exp\left(-\frac{d_r^+(u)\lceil f(n_r) \rceil}{n_r}\right) d_r^+(u)$$

$$\leq \max_{z > 0} \exp\left(-\frac{z \lceil f(n_r) \rceil}{n_r}\right) z$$

$$\leq \frac{n_r}{e\lceil f(n_r) \rceil}$$

$$\leq \frac{n_r}{e f(n_r)} \,.$$

Combining with the above, this implies

$$\sum_{e \in \mathcal{E}} \mathbb{P}(B_2(e)) \leq \frac{1}{2e} \sum_{r=1}^{\lceil f(n-1) \rceil} \mathbb{E}\left[\frac{n_r}{f(n_r)}\right] \leq \frac{1}{2e} \sum_{r=1}^{\lceil f(n-1) \rceil} \frac{n}{f(n)} \leq \frac{n}{e}$$

where we used the facts that $n_r \leq n$ and the properties of $f$.

**Bounding $\sum_{e \in \mathcal{E}} \mathbb{P}(B_3(e))$.** Let $V_{\text{fin}}$ be the remaining vertices in $V_r$ after the algorithm stops and assume $|V_{\text{fin}}| > 1$ (so that there is at least a query left). Let $n_{\text{fin}} = |V_{\text{fin}}| - 1$ and, for any $u \in V_{\text{fin}}$, let $d_{\text{fin}}^+(u) = \big| \{v \in V_{\text{fin}} \,:\, \sigma(u,v) = +1\} \big|$. In what follows, we conventionally assume $V_r \equiv V_{\text{fin}}$ for any $r > \lceil f(n-1) \rceil$, and similarly for $n_{\text{fin}}$ and $d_{\text{fin}}^+$. We have

$$\sum_{e \in \mathcal{E}} \mathbb{I}\{B_3(e)\} = \frac{1}{2} \sum_{u \in V_{\text{fin}}} d_{\text{fin}}^+(u) \leq \frac{1}{2}\left(\sum_{u \in V_{\text{fin}}} \frac{n_{\text{fin}}}{\lceil f(n_{\text{fin}}) \rceil} + \sum_{u \in V_{\text{fin}}} \mathbb{I}\left\{d_{\text{fin}}^+(u) > \frac{n_{\text{fin}}}{\lceil f(n_{\text{fin}}) \rceil}\right\} d_{\text{fin}}^+(u)\right) \,.$$

Fix some $r \leq \lceil f(n-1) \rceil$. Given any vertex $v \in V_r$ with $d_r^+(v) \geq \frac{n_r}{\lceil f(n_r) \rceil}$, let $E_r(v)$ be the event that, at round $r$, ACC queries $\sigma(v,u)$ for all $u \in V_r \setminus \{v\}$. Introduce the notation $S_r = \sum_{u \in V_r} \mathbb{I}\left\{ d_r^+(u) > \frac{n_r}{\lceil f(n_r) \rceil} \right\} d_r^+(u)$ with $S_r = S_{\text{fin}}$ for all $r > \lceil f(n) \rceil$, and let $\delta_r = n_r - n_{r+1}$ be the number of nodes that are removed from $V_r$ at the end of the $r$-th recursive call. Then

$$\delta_r \geq \mathbb{I}\{E_r(\pi_r)\} d_r^+(\pi_r) \geq \mathbb{I}\left\{ d_r^+(\pi_r) > \frac{n_r}{\lceil f(n_r) \rceil} \right\} \mathbb{I}\{E_r(\pi_r)\} d_r^+(\pi_r)$$

and

$$\mathbb{E}[\delta_r \mid H_{r-1}] \geq \sum_{v \in V_r} \mathbb{I}\left\{ d_r^+(v) > \frac{n_r}{\lceil f(n_r) \rceil} \right\} \mathbb{P}\big(E_r(v) \mid \pi_r = v, H_{r-1}\big) \mathbb{P}(\pi_r = v \mid H_{r-1}) d_r^+(v).$$

Using the same argument as the one we used to bound (6),

$$\mathbb{P}\big(E_r(v) \mid \pi_r = v, H_{r-1}\big) \geq 1 - \left(1 - \frac{d_r^+(v)}{n_r}\right)^{\lceil f(n_r) \rceil} \geq 1 - \left(1 - \frac{1}{\lceil f(n_r) \rceil}\right)^{\lceil f(n_r) \rceil} \geq 1 - \frac{1}{e}$$

and $\mathbb{P}(\pi_r = v \mid H_{r-1}) = \frac{1}{n_r + 1}$ for any $v \in V_r$, we may write

$$\mathbb{E}[\delta_r \mid H_{r-1}] \geq \left(1 - \frac{1}{e}\right) \frac{\mathbb{E}[S_r \mid H_{r-1}]}{n_r + 1} \geq \left(1 - \frac{1}{e}\right) \frac{\mathbb{E}[S_r \mid H_{r-1}]}{n}.$$

Observe now that $\sum_{r=1}^{\lceil f(n-1) \rceil} \delta_r \leq n_1 - n_{\text{fin}} \leq n - 1$ and $S_r$ is monotonically nonincreasing in $r$. Thus

$$n - 1 \geq \sum_{r=1}^{\lceil f(n-1) \rceil} \mathbb{E}[\delta_r] \geq \frac{1}{n} \left(1 - \frac{1}{e}\right) \sum_{r=1}^{\lceil f(n) \rceil} \mathbb{E}[S_r] \geq \frac{\lceil f(n-1) \rceil}{n} \left(1 - \frac{1}{e}\right) \mathbb{E}[S_{\text{fin}}]$$

which implies $\mathbb{E}[S_{\text{fin}}] \leq \left(\frac{e}{e-1}\right) \frac{n(n-1)}{\lceil f(n-1) \rceil} \leq \left(\frac{e}{e-1}\right) \frac{n(n-1)}{f(n-1)}$. By the properties of $f$, however, $\left(\frac{e}{e-1}\right) \frac{n(n-1)}{f(n-1)} \leq \left(\frac{e}{e-1}\right) \frac{n^2}{f(n)}$. So we have

$$\sum_{e \in \mathcal{E}} \mathbb{P}(B_3(e)) \leq \frac{1}{2} \left( \sum_{u \in V_{\text{fin}}} \mathbb{E}\left[\frac{n_{\text{fin}}}{f(n_{\text{fin}})}\right] + \mathbb{E}[S_{\text{fin}}] \right) \leq \frac{1}{2} \left( \frac{n^2}{f(n)} + \frac{e}{e-1} \frac{n^2}{f(n)} \right)$$

as claimed.

**Bounding the number of queries.**  In any given round, ACC asks less than $n$ queries. Since the number of rounds is at most $\lceil f(n) \rceil$, the overall number of queries is less than $n \lceil f(n) \rceil$.

**KwikCluster as special case.**  One can immediately see that, if $f(n) = n$ for all $n$, then ACC coincides with KwikCluster and therefore the bound $\mathbb{E}[\Delta] \leq 3\text{OPT}$ applies [1].

## 2.3  Pseudocode of ACCESS

## 2.4  Proof of Theorem 2

We refer to the pseudocode of ACCESS (Algorithm 2).

**Bounding $\mathbb{E}[\Delta_A]$.**  Let $G_r$ be the residual graph at round $r$. The total clustering cost $\Delta_A$ of ACCESS can be bounded by the sum of two terms: the clustering cost $\Delta_1$ of ACC without round restriction (i.e. ACC terminating only when the residual graph is empty), and the number of edges $\Delta_2$ in the residual graph $G_r$ if $r$ is the round at which ACCESS stops. Concerning $\Delta_1$, the proof of Theorem 1 shows that $\mathbb{E}[\Delta_1] \leq 3\text{OPT} + n/e$. Concerning $\Delta_2$, we have two cases. If ACCESS stops at line 1, then obviously $\Delta_2 \leq 2n^2/f(n)$. If instead ACCESS stops at line 4, then note that for any $k \geq 0$ the probability that such an event happens given that $\Delta_2 = k$ is at most:

$$\left(1 - \frac{k}{\binom{|V_r|}{2}}\right)^{\left\lceil \binom{|V_r|}{2} f(n)/n^2 \right\rceil} \leq e^{-kf(n)/n^2}$$

Thus $\mathbb{E}[\Delta_2] \leq \max_{k \geq 1}\left(k e^{-kf(n)/n^2}\right) \leq \frac{n^2}{ef(n)} < 2n^2/f(n)$.

---
**Algorithm 2** ACCESS with query rate $f$
---
**Parameters:** residual node set $V_r$, round index $r$

1: **if** $\binom{|V_r|}{2} \le 2n^2/f(n)$ **then** STOP and declare every $v \in V_r$ as singleton
2: Sample the labels of $\lceil \binom{|V_r|}{2} f(n)/n^2 \rceil$ pairs chosen u.a.r. from $\binom{V_r}{2}$
3: **if** no label is positive **then**
4:     STOP and declare every $v \in V_r$ as singleton
5: Draw pivot $\pi_r$ u.a.r. from $V_r$
6: $C_r \leftarrow \{\pi_r\}$                 ▷ Create new cluster and add the pivot to it
7: Draw a random subset $S_r$ of $\lceil f(|V_r| - 1) \rceil$ nodes from $V_r \setminus \{\pi_r\}$
8: **for** each $u \in S_r$ **do** query $\sigma(\pi_r, u)$
9: **if** $\exists\, u \in S_r$ such that $\sigma(\pi_r, u) = +1$ **then**     ▷ Check if there is at least an edge
10:     Query all remaining pairs $(\pi_r, u)$ for $u \in V_r \setminus \big(\{\pi_r\} \cup S_r\big)$
11:     $C_r \leftarrow C_r \cup \{u\,:\,\sigma(\pi_r, u) = +1\}$     ▷ Populate cluster based on queries
12: Output cluster $C_r$
13: ACCESS$(V_r \setminus C_r, r+1)$              ▷ Recursive call on the remaining nodes

---

**Bounding** $\mathbb{E}[Q]$. The queries performed at line 1 are deterministically at most $n\lceil f(n) \rceil$. Concerning the other queries (line 8 and line 10), we divide the algorithm in two phases: the "heavy" rounds $r$ where $G_r$ still contains at least $n^2/(2f(n))$ edges, and the remaining "light" rounds where $G_r$ contains less than $n^2/(2f(n))$ edges.

Consider first a "heavy" round $r$. We see $G_r$ as an arbitrary fixed graph: for all random variables mentioned below, the distribution is thought solely as a function of the choices of the algorithm in the current round (i.e., the pivot node $\pi_r$ and the queried edges). Now, let $Q_r$ be the number of queries performed at lines 8 and 10), and $R_r = |V_r| - |V_{r+1}|$ be the number of nodes removed. Let $\pi_r$ be the pivot, and let $D_r$ be its degree in $G_r$. Let $X_r$ be the indicator random variable of the event that $\sigma(\pi_r, u) = +1$ for some $u \in S_r$. Observe that:

$$Q_r \le \lceil f(|V_r| - 1) \rceil + X_r(|V_r| - 1) \qquad \text{and} \qquad R_r = 1 + X_r D_r$$

Thus $\mathbb{E}[Q_r] \le \lceil f(|V_r| - 1) \rceil + \mathbb{E}[X_r]|V_r|$, while $\mathbb{E}[R_r] = 1 + \mathbb{E}[X_r D_r]$. However, $X_r$ is monotonically increasing in $D_r$, so $\mathbb{E}[X_r D_r] = \mathbb{E}[X_r]\mathbb{E}[D_r] + \mathrm{Cov}(X_r, D_r) \ge \mathbb{E}[X_r]\mathbb{E}[D_r]$. Moreover, by hypothesis $\mathbb{E}[D_r] \ge 2\big(n^2/(2f(n))\big)/|V_r| \ge n/f(n)$. Thus:

$$\mathbb{E}[R_r] \ge 1 + \mathbb{E}[X_r]\mathbb{E}[D_r]$$
$$\ge 1 + \mathbb{E}[X_r]\frac{n}{f(n)}$$
$$\ge 1 + \mathbb{E}[X_r]\frac{|V_r|}{f(|V_r|)}$$
$$\ge 1 + \mathbb{E}[X_r]\frac{|V_r|}{\lceil f(|V_r|) \rceil}$$
$$\ge \frac{\mathbb{E}[Q_r]}{\lceil f(|V_r|) \rceil}$$
$$\ge \frac{\mathbb{E}[Q_r]}{\lceil f(n) \rceil}$$

But then, since obviously $\sum_r R_r \le n$:

$$\mathbb{E}\left[\sum_{r \text{ heavy}} Q_r\right] \le \lceil f(n) \rceil \mathbb{E}\left[\sum_{r \text{ heavy}} R_r\right] \le n\lceil f(n) \rceil$$

Consider now the "light" rounds, where $G_r$ contains less than $n^2/(2f(n))$ edges. In any such round the expected number of edges found at line 2 is less than:

$$\frac{n^2/f(n)}{2\binom{|V_r|}{2}}\left\lceil \binom{|V_r|}{2} f(n)/n^2 \right\rceil \tag{7}$$

However, $\binom{|V_r|}{2} > 2n^2/f(n)$ otherwise ACCESS would have stopped at line 1, hence:

$$\left\lceil \binom{|V_r|}{2} f(n)/n^2 \right\rceil \leq \frac{3}{2}\binom{|V_r|}{2} f(n)/n^2 \tag{8}$$

which implies that the expression in (7) is bounded by $\frac{3}{4}$. By Markov's inequality this is also an upper bound on the probability that ACCESS finds some edge at line 2, so in every light round ACCESS stops at line 4 with probability at least $\frac{1}{4}$. Hence ACCESS completes at most 4 light rounds in expectation; the corresponding expected number of queries is then at most $4n$.

## 2.5 Proof of Theorem 3

First of all, note that if the residual graph $G_r$ contains $\mathcal{O}(n^2/f(n))$ edges, from $r$ onward ACCESS stops at each round independently with constant probability. The expected number of queries performed before stopping is therefore $\mathcal{O}(n)$, and the expected error incurred is obviously at most $\mathcal{O}(n^2/f(n))$.

We shall then bound the expected number of queries required before the residual graph contains $\mathcal{O}(n^2/f(n))$ edges. In fact, by definition of $i'$, if ACCESS removes $C_{i'}, \ldots, C_\ell$, then the residual graph contains $\mathcal{O}(n^2/f(n))$ edges. We therefore bound the expected number of queries before $C_{i'}, \ldots, C_\ell$ are removed.

First of all recall that, when pivoting on a cluster of size $c$, the probability that the cluster is *not* removed is at most $e^{-cf(n)/n}$. Thus the probability that the cluster is not removed after $\Omega(c)$ of its nodes have been used as pivot is $e^{-\Omega(c^2)f(n)/n}$. Hence the probability that *any* of $C_{i'}, \ldots, C_\ell$ is not removed after $\Omega(c)$ of its nodes are used as pivot is, setting $c = \Omega(h(n))$ and using a union bound, at most $p = ne^{-\Omega(h(n)^2)f(n)/n}$. Observe that $h(n) = \Omega(n/f(n))$, for otherwise $\sum_{j=1}^{i'} \binom{C_j}{2} = o(n^2/f(n))$, a contradiction. Therefore $p \leq ne^{-\Omega(h(n))}$. Note also that we can assume $h(n) = \omega(\ln n)$, else the theorem bound is trivially $O(n^2)$. This gives $p = \mathcal{O}(ne^{-\omega(\ln n)}) = o(1/\operatorname{poly}(n))$. We can thus condition on the events that, at any point along the algorithm, every cluster among $C_{i'}, \ldots, C_\ell$ that is still in the residual graph has size $\Omega(h(n))$; the probability of any other event changes by an additive $\mathcal{O}(p)$, which can be ignored.

Let now $k = \ell - i' + 1$, and suppose at a generic point $k' \leq k$ of the clusters $C_{i'}, \ldots, C_\ell$ are in the residual graph. Their total size is therefore $\Omega(k'h(n))$. Therefore $\mathcal{O}(n/k'h(n))$ rounds in expectation are needed for the pivot to fall among those clusters. Each time this happens, with probability $1 - e^{-\Omega(h(n))f(n)/n} = \Omega(1)$ the cluster containing the pivot is removed. Hence, in expectation a new cluster among $C_{i'}, \ldots, C_\ell$ is removed after $\mathcal{O}(n/k'h(n))$ rounds. By summing over all values of $k'$, the number of expected rounds to remove all of $C_{i'}, \ldots, C_\ell$ is

$$\mathcal{O}\left(\sum_{k'=1}^{k} \frac{n}{k'h(n)}\right) = \mathcal{O}(n(\ln n)/h(n))$$

Since each round involves $\mathcal{O}(n)$ queries, the bound follows.

# 3 Supplementary Material for Section 4

## 3.1 Proof of Theorem 4

Fix any $C$ that is $(1 - \varepsilon)$-knit. We show that ACC outputs a $\widehat{C}$ such that

$$\mathbb{E}\big[|\widehat{C} \cap C|\big] \geq \max\left\{\left(1 - \frac{5}{2}\varepsilon\right)|C| - 2\frac{n}{f(n)}, \left(\frac{f(n)}{n} - \frac{5}{2}\varepsilon\right)|C|\right\} \text{ and } \mathbb{E}\big[|\widehat{C} \cap \overline{C}|\big] \leq \frac{\varepsilon}{2}|C| \tag{9}$$

One can check that these two conditions together imply the first two terms in the bound. We start by deriving a lower bound on $\mathbb{E}\big[|\widehat{C} \cap C|\big]$ for KwikCluster assuming $|E_C| = \binom{|C|}{2}$. Along the way we introduce most of the technical machinery. We then port the bound to ACC, relax the assumption to $|E_C| \geq (1 - \varepsilon)\binom{|C|}{2}$, and bound $\mathbb{E}\big[|\widehat{C} \cap \overline{C}|\big]$ from above. Finally, we add the $|C|e^{-|C|f(n)/5n}$ part of the bound. To lighten the notation, from now on $C$ denotes both the cluster and its cardinality $|C|$.

For the sake of analysis, we see KwikCluster as the following equivalent process. First, we draw a random permutation $\pi$ of $V$. This is the ordered sequence of *candidate pivots*. Then, we set $G_1 = G$, and for each $i = 1, \ldots, n$ we proceed as follows. If $\pi_i \in G_i$, then $\pi_i$ is used as an actual pivot; in this case we let $G_{i+1} = G_i \setminus (\pi_i \cup \mathcal{N}_{\pi_i})$ where $\mathcal{N}_v$ is the set of neighbors of $v$. If instead $\pi_i \notin G_i$, then we let $G_{i+1} = G_i$. Hence, $G_i$ is the residual graph just before the $i$-th candidate pivot $\pi_i$ is processed. We indicate the event $\pi_i \in G_i$ by the random variable $P_i$:

$$P_i = \mathbb{I}\{\pi_i \in G_i\} = \mathbb{I}\{\pi_i \text{ is used as pivot}\} \tag{10}$$

More in general, we define a random variable indicating whether node $v$ is "alive" in $G_i$:

$$X(v, i) = \mathbb{I}\{v \in G_i\} = \mathbb{I}\left\{v \notin \cup_{j < i \,:\, P_j = 1} (\pi_j \cup \mathcal{N}_{\pi_j})\right\} \tag{11}$$

Let $i_C = \min\{i : \pi_i \in C\}$ be the index of the first candidate pivot of $C$. Define the random variable:

$$S_C = |C \cap G_{i_C}| = \sum_{v \in C} X(v, i_C) \tag{12}$$

In words, $S_C$ counts the nodes of $C$ still alive in $G_{i_C}$. Now consider the following random variable:

$$S = P_{i_C} \cdot S_C \tag{13}$$

Let $\widehat{C}$ be the cluster that contains $\pi_{i_C}$ in the output of KwikCluster. It is easy to see that $|C \cap \widehat{C}| \geq S$. Indeed, if $P_{i_C} = 1$ then $\widehat{C}$ includes $C \cap G_{i_C}$, so $|C \cap \widehat{C}| \geq P_{i_C} S_C = S$. If instead $P_{i_C} = 0$, then $S = 0$ and obviously $|C \cap \widehat{C}| \geq 0$. Hence in any case $|C \cap \widehat{C}| \geq S$, and $\mathbb{E}[|C \cap \widehat{C}|] \geq \mathbb{E}[S]$. Therefore we can bound $\mathbb{E}[|C \cap \widehat{C}|]$ from below by bounding $\mathbb{E}[S]$ from below.

Before continuing, we simplify the analysis by assuming KwikCluster runs on the graph $G$ after all edges not incident on $C$ have been deleted. We can easily show that this does not increase $S$. First, by (11) each $X(v, i_C)$ is a nonincreasing function of $\{P_i : i < i_C\}$. Second, by (12) and (13), $S$ is a nondecreasing function of $\{X(v, i_C) : v \in C\}$. Hence, $S$ is a nonincreasing function of $\{P_i : i < i_C\}$. Now, the edge deletion forces $P_i = 1$ for all $i < i_C$, since any $\pi_i : i < i_C$ has no neighbor $\pi_j : j < i$. Thus the edge deletion does not increase $S$ (and, obviously, $\mathbb{E}[S]$). We can then assume $G[V \setminus C]$ is an independent set. At this point, any node not adjacent to $C$ is isolated and can be ignored. We can thus restrict the analysis to $C$ and its neighborhood in $G$. Therefore we let $\overline{C} = \{v : \{u, v\} \in E, u \in C, v \notin C\}$ denote both the neighborhood and the complement of $C$.

We turn to bounding $\mathbb{E}[S]$. For now we assume $G[C]$ is a clique; we will then relax the assumption to $|E_C| \geq (1 - \varepsilon)\binom{C}{2}$. Since by hypothesis $\mathrm{cut}(C, \overline{C}) < \varepsilon C^2$, the average degree of the nodes in $\overline{C}$ is less than $\varepsilon C^2 / \overline{C}$. This is also a bound on the expected number of edges between $C$ and a node drawn u.a.r. from $\overline{C}$. But, for any given $i$, conditioned on $i_C - 1 = i$ the nodes $\pi_1, \ldots, \pi_{i_C - 1}$ are indeed drawn u.a.r. from $\overline{C}$, and so have a total of at most $i\varepsilon C^2/\overline{C}$ edges towards $C$ in expectation. Thus, over the distribution of $\pi$, the expected number of edges between $C$ and $\pi_1, \ldots, \pi_{i_C - 1}$ is at most:

$$\sum_{i=0}^{n} \frac{i\varepsilon C^2}{\overline{C}} \mathbb{P}(i_C - 1 = i) = \frac{\varepsilon C^2}{\overline{C}} \mathbb{E}[i_C - 1] = \frac{\varepsilon C^2}{\overline{C}} \frac{\overline{C}}{C + 1} < \varepsilon C \tag{14}$$

where we used the fact that $\mathbb{E}[i_C - 1] = \overline{C}/(C + 1)$. Now note that (14) is a bound on $C - \mathbb{E}[S_C]$, the expected number of nodes of $C$ that are adjacent to $\pi_1, \ldots, \pi_{i_C - 1}$. Therefore, $\mathbb{E}[S_C] \geq (1 - \varepsilon)C$.

Recall that $P_{i_C}$ indicates whether $\pi_{i_C}$ is not adjacent to any of $\pi_1, \ldots, \pi_{i_C - 1}$. Since the distribution of $\pi_{i_C}$ is uniform over $C$, $\mathbb{P}(P_{i_C} \mid S_C) = S_C/C$. But $S = P_{i_C} S_C$, hence $\mathbb{E}[S \mid S_C] = (S_C)^2/C$, and thus $\mathbb{E}[S] = \mathbb{E}[(S_C)^2]/C$. Using $\mathbb{E}[S_C] \geq (1 - \varepsilon)C$ and invoking Jensen's inequality we obtain

$$\mathbb{E}[S] \geq \frac{\mathbb{E}[S_C]^2}{C} \geq (1 - \varepsilon)^2 C \geq (1 - 2\varepsilon)C \tag{15}$$

which is our bound on $\mathbb{E}[|C \cap \widehat{C}|]$ for KwikCluster.

Let us now move to ACC. We have to take into account the facts that ACC performs $f(|G_r| - 1)$ queries on the pivot before deciding whether to perform $|G_r| - 1$ queries, and that ACC stops after $f(n - 1)$ rounds. We start by addressing the first issue, assuming for the moment ACC has no restriction on the number of rounds.

Recall that $\mathbb{P}(P_{i_C} \mid S_C) = S_C/C$. Now, if $P_{i_C} = 1$, then we have $S_C - 1$ edges incident on $\pi_{i_C}$. It is easy to check that, if $n_r + 1$ is the number of nodes at the round when $\pi_{i_C}$ is used, then the probability that ACC finds some edge incident on $\pi_{i_C}$ is at least:

$$1 - \left(1 - \frac{S_C - 1}{n_r}\right)^{\lceil f(n_r) \rceil} \geq 1 - e^{-f(n_r)\frac{S_C-1}{n_r}} \geq 1 - e^{-f(n)\frac{S_C-1}{n}} \tag{16}$$

and, if this event occurs, then $S = S_C$. Thus

$$\mathbb{E}[S \mid S_C] = \mathbb{P}(P_{i_C} \mid S_C)S_C \geq \left(1 - e^{-f(n)\frac{S_C-1}{n}}\right)\frac{S_C^2}{C} \geq \frac{S_C^2}{C} - S_C \frac{2n}{f(n)C} \tag{17}$$

where we used the facts that for $S_C \leq 1$ the middle expression in (17) vanishes, that $e^{-x} < 1/x$ for $x > 0$, and that $1/x < 2/(x+1)$ for all $x \geq 2$. Simple manipulations, followed by Jensen's inequality and an application of $\mathbb{E}[S_C] \geq (1 - \varepsilon)C$, give

$$\mathbb{E}[S] \geq (1 - \varepsilon)^2 C - (1 - \varepsilon)C\frac{2n}{f(n)C} \geq (1 - 2\varepsilon)C - 2\frac{n}{f(n)} \tag{18}$$

We next generalize the bound to the case $E_C \geq (1 - \varepsilon)\binom{C}{2}$. To this end note that, since at most $\varepsilon\binom{C}{2}$ edges are missing from any subset of $C$, then any subset of $S_C$ nodes of $C$ has average degree at least

$$\max\left\{0, S_C - 1 - \binom{C}{2}\frac{2\varepsilon}{S_C}\right\} \geq S_C - \frac{\varepsilon C(C-1)}{2S_C} - 1 \tag{19}$$

We can thus re-write (17) as

$$\mathbb{E}[S \mid S_C] \geq \frac{S_C}{C}\left(1 - e^{-f(n)\frac{S_C-1}{n}}\right)\left(S_C - \frac{\varepsilon C(C-1)}{2S_C}\right) \tag{20}$$

Standard calculations show that this expression is bounded from below by $\frac{S_C^2}{C} - S_C\frac{2n}{f(n)C} - \frac{\varepsilon C}{2}$, which by calculations akin to the ones above leads to $\mathbb{E}[S] \geq (1 - \frac{5}{2}\varepsilon)C - 2\frac{n}{f(n)}$.

Similarly, we can show that $\mathbb{E}[S] \geq \left(\frac{f(n)}{n} - \frac{5}{2}\varepsilon\right)C$. To this end note that when ACC pivots on $\pi_{i_C}$ all the remaining cluster nodes are found with probability at least $\frac{f(n)}{n}$ (this includes the cases $S_C \leq 1$, when such a probability is indeed 1). In (17), we can then replace $1 - e^{-f(n)\frac{S_C-1}{n}}$ with $\frac{f(n)}{n}$, which leads to $\mathbb{E}[S] \geq \left(\frac{f(n)}{n} - \frac{5}{2}\varepsilon\right)C$. This proves the first inequality in (9).

For the second inequality in (9), note that any subset of $S_C$ nodes has $\text{cut}(C, \overline{C}) \leq \varepsilon\binom{C}{2}$. Thus, $\pi_{i_C}$ is be incident to at most $\frac{\varepsilon}{S_C}\binom{C}{2}$ such edges in expectation. The expected number of nodes of $\overline{C}$ that ACC assigns to $\widehat{C}$, as a function of $S_C$, can thus be bounded by $\frac{S_C}{C}\frac{\varepsilon}{S_C}\binom{C}{2} < \frac{\varepsilon}{2}C$.

As far as the $\mathcal{O}(Ce^{-Cf(n)/n})$ part of the bound is concerned, simply note that the bounds obtained so far hold unless $i_C > \lceil f(n-1) \rceil$, in which case ACC stops before ever reaching the first node of $C$. If this happens, $\widehat{C} = \{\pi_{i_C}\}$ and $|\widehat{C} \oplus C| < |C|$. The event $i_C > \lceil f(n-1) \rceil$ is the event that no node of $C$ is drawn when sampling $\lceil f(n-1) \rceil$ nodes from $V$ without replacement. We can therefore apply Chernoff-type bounds to the random variable $X$ counting the number of draws of nodes of $C$ and get $\mathbb{P}(X < (1-\beta)\mathbb{E}[X]) \leq \exp(-\beta^2\mathbb{E}[X]/2)$ for all $\beta > 0$. In our case $\mathbb{E}[X] = \lceil f(n-1) \rceil|C|/n$, and we have to bound the probability that $X$ equals $0 < (1 - \beta)\mathbb{E}[X]$. Thus

$$\mathbb{P}(X = 0) \leq \exp\left(-\frac{\beta^2\mathbb{E}[X]}{2}\right) = \exp\left(-\frac{\beta^2\lceil f(n-1)\rceil|C|}{2n}\right)$$

Note however that $\lceil f(n-1) \rceil \geq f(n)/2$ unless $n = 1$ (in which case $V$ is trivial). Then, choosing e.g. $\beta > \sqrt{4/5}$ yields $\mathbb{P}(X = 0) < \exp\left(-|C|f(n)/5n\right)$. This case therefore adds at most $|C|\exp(-|C|f(n)/5n)$ to $\mathbb{E}[|\widehat{C} \oplus C|]$.

## 3.2   Proof of Theorem 5

Before moving to the actual proof, we need some ancillary results. The next lemma bounds the probability that ACC does not pivot on a node of $C$ in the first $k$ rounds.

**Lemma 2.** *Fix a subset $C \subseteq V$ and an integer $k \geq 1$, and let $\pi_1, \ldots, \pi_n$ be a random permutation of $V$. For any $v \in C$ let $X_v = \mathbb{I}\{v \in \{\pi_1, \ldots, \pi_k\}\}$, and let $X_C = \sum_{v \in C} X_v$. Then $\mathbb{E}[X_C] = \frac{k|C|}{n}$, and $\mathbb{P}(X_C = 0) < e^{-\frac{k|C|}{3n}}$.*

*Proof.* Since $\pi$ is a random permutation, then for each $v \in C$ and each each $i = 1, \ldots, k$ we have $\mathbb{P}(\pi_i = v) = \frac{1}{n}$. Therefore $\mathbb{E}[X_v] = \frac{k}{n}$ and $\mathbb{E}[X_C] = \frac{k|C|}{n}$. Now, the process is exactly equivalent to sampling without replacement from a set of $n$ items of which $|C|$ are marked. Therefore, the $X_v$'s are non-positively correlated and we can apply standard concentration bounds for the sum of independent binary random variables. In particular, for any $\eta \in (0,1)$ we have:

$$\mathbb{P}(X_C = 0) \leq \mathbb{P}(X_C < (1-\eta)\mathbb{E}[X_C]) < \exp\left(-\frac{\eta^2 \mathbb{E}[X_C]}{2}\right)$$

which drops below $e^{-\frac{k|C|}{3n}}$ by replacing $\mathbb{E}[X_C]$ and choosing $\eta \geq \sqrt{2/3}$. □

The next lemma is the crucial one.

**Lemma 3.** *Let $\varepsilon \leq \frac{1}{10}$. Consider a strongly $(1-\varepsilon)$-knit set $C$ with $|C| > \frac{10n}{f(n)}$. Let $u_C = \min\{v \in C\}$ be the id of $C$. Then, for any $v \in C$, in any single run of ACC we have $\mathbb{P}(\mathrm{id}(v) = u_C) \geq \frac{2}{3}$.*

*Proof.* We bound from above the probability that any of three "bad" events occurs. As in the proof of Theorem 4, we equivalently see ACC as going through a sequence of candidate pivots $\pi_1, \ldots, \pi_n$ that is a uniform random permutation of $V$. Let $i_C = \min\{i : \pi_i \in C\}$ be the index of the first node of $C$ in the random permutation of candidate pivots. The first event, $B_1$, is $\{i_C > \lceil f(n-1) \rceil\}$. Note that, if $B_1$ does not occur, then ACC will pivot on $\pi_{i_C}$. The second event, $B_2$, is the event that $\pi_{i_C} \in V_{sing}$ if ACC pivots on $\pi_{i_C}$ (we measure the probability of $B_2$ conditioned on $\overline{B_1}$). The third event, $B_3$, is $\{\pi_{i_C} \notin P\}$ where $P = \mathcal{N}_{u_C} \cap \mathcal{N}_v$. If none among $B_1, B_2, B_3$ occurs, then ACC forms a cluster $\widehat{C}$ containing both $u_C$ and $v$, and by the min-tagging rule sets $\mathrm{id}(v) = \min_{u \in \widehat{C}} = u_C$. We shall then show that $\mathbb{P}(B_1 \cup B_2 \cup B_3) \leq 1/3$.

For $B_1$, we apply Lemma 2 by observing that $i_C > \lceil f(n-1) \rceil$ corresponds to the event $X_C = 0$ with $k = \lceil f(n-1) \rceil$. Thus

$$\mathbb{P}(i_C > \lceil f(n-1) \rceil) < e^{-\frac{\lceil f(n-1) \rceil |C|}{3n}} \leq e^{-\frac{f(n-1)}{3n}\frac{10n}{f(n)}} = e^{-\frac{f(n-1)}{f(n)}\frac{10}{3}} < e^{-3}$$

where we used the fact that $n \geq |C| \geq 11$ and therefore $f(n-1) \geq \frac{10}{11}f(n)$.

For $B_2$, recall that by definition every $v \in C$ has at least $(1-\varepsilon)c$ edges. By the same calculations as the ones above, if ACC pivots on $\pi_{i_C}$, then:

$$\mathbb{P}(\pi_{i_C} \in V_{sing}) \leq \exp\left(-\frac{f(n-1)}{n-1}(1-\varepsilon)c\right) \leq \exp\left(-\frac{f(n-1)}{n-1}\left(1-\frac{1}{10}\right)\frac{10n}{f(n)}\right) \leq e^{-9}$$

For $B_3$, note that the distribution of $\pi_{i_C}$ is uniform over $C$. Now, let $\mathcal{N}_{u_C}$ and $\mathcal{N}_v$ be the neighbor sets of $u_C$ and $v$ in $C$, and let $P = \mathcal{N}_{u_C} \cap \mathcal{N}_v$. We call $P$ the set of good pivots. Since $C$ is strongly $(1-\varepsilon)$-knit, both $u_C$ and $v$ have at least $(1-\varepsilon)c$ neighbors in $C$. But then $|C \setminus P| \leq 2\varepsilon c$ and

$$\mathbb{P}(\pi_{i_C} \notin P) = \frac{|C \setminus P|}{|C|} \leq 2\varepsilon \leq 1/5$$

By a union bound, then, $\mathbb{P}(B_1 \cup B_2 \cup B_3) \leq e^{-3} + e^{-9} + 1/5 < 1/3$. □

We are now ready to conclude the proof. Suppose we execute ACC independently $K = 48\lceil \ln(n/p) \rceil$ times with the min-tagging rule. For a fixed $v \in G$ let $X_v$ be the number of executions giving $\mathrm{id}(v) = u_C$. On the one hand, by Lemma 3, $\mathbb{E}[X_v] \geq \frac{2}{3}K$. On the other hand, $v$ will not be assigned to the cluster with id $u_C$ by the majority voting rule only if $X_v \leq \frac{1}{2}K \leq \mathbb{E}[X_v](1-\delta)$ where $\delta = \frac{1}{4}$. By standard concentration bounds, then, $\mathbb{P}(X_v \leq \frac{1}{2}K) \leq \exp(-\frac{\delta^2 \mathbb{E}[X_v]}{2}) = \exp(-\frac{K}{48})$. By setting $K = 48\ln(n/p)$, the probability that $v$ is not assigned id $u_C$ is thus at most $p/n$. A union bound over all nodes concludes the proof.

# 4 Supplementary Material for Section 6

## 4.1 Proof of Theorem 8

We prove that there exists a distribution over labelings $\sigma$ with $\mathrm{OPT} = 0$ on which any deterministic algorithm has expected cost at least $\frac{n\varepsilon^2}{8}$. Yao's minimax principle then implies the claimed result.

Given $V = \{1, \ldots, n\}$, we define $\sigma$ by a random partition of the vertices in $d \geq 2$ isolated cliques $T_1, \ldots, T_d$ such that $\sigma(v, v') = +1$ if and only if $v$ and $v'$ belong to the same clique. The cliques are formed by assigning each node $v \in V$ to a clique $I_v$ drawn uniformly at random with replacement from $\{1, \ldots, d\}$, so that $T_i = \{v \in V : I_v = i\}$. Consider a deterministic algorithm making queries $\{s_t, r_t\} \in \mathcal{E}$. Let $E_i$ be the event that the algorithm never queries a pair of nodes in $T_i$ with $|T_i| \geq \frac{n}{2d} > 5$. Apply Lemma 4 below with $d = \frac{1}{\varepsilon}$. This implies that the expected number of non-queried clusters of size at least $\frac{n}{2d}$ is at least $\frac{d}{2} = \frac{1}{2\varepsilon}$. The overall expected cost of ignoring these clusters is therefore at least

$$\frac{d}{2}\left(\frac{n}{2d}\right)^2 = \frac{n^2}{8d} = \frac{\varepsilon n^2}{8}$$

and this concludes the proof.

**Lemma 4.** *Suppose $d > 0$ is even, $n \geq 16d \ln d$, and $B < \frac{d^2}{50}$. Then for any deterministic learning algorithm making at most $B$ queries,*

$$\sum_{i=1}^{d} \mathbb{P}(E_i) > \frac{d}{2} .$$

*Proof.* For each query $\{s_t, r_t\}$ we define the set $L_t$ of all cliques $T_i$ such that $s_t \notin T_i$ and some edge containing both $s_t$ and a node of $T_i$ was previously queried. The set $R_t$ is defined similarly using $r_t$. Formally,

$$L_t = \{i : (\exists \tau < t)\; s_\tau = s_t \wedge r_\tau \in T_i \wedge \sigma(s_\tau, r_\tau) = -1\}$$
$$R_t = \{i : (\exists \tau < t)\; r_\tau = r_t \wedge s_\tau \in T_i \wedge \sigma(s_\tau, r_\tau) = -1\} .$$

Let $D_t$ be the event that the $t$-th query discovers a new clique of size at least $\frac{n}{2d}$, and let $P_t = \max\{|L_t|, |R_t|\}$. Using this notation,

$$\sum_{t=1}^{B} \mathbb{I}\{D_t\} = \sum_{t=1}^{B} \mathbb{I}\{D_t \wedge P_t < d/2\} + \underbrace{\sum_{t=1}^{B} \mathbb{I}\{D_t \wedge P_t \geq d/2\}}_{N} . \tag{21}$$

We will now show that unless $B \geq \frac{d^2}{50}$, we can upper bound $N$ deterministically by $\sqrt{2B}$.

Suppose $N > \frac{d}{2}$, and let $t_1, \ldots, t_N$ be the times $t_k$ such that $\mathbb{I}\{D_{t_k} \wedge P_{t_k} \geq d/2\} = 1$. Now fix some $k$ and note that, because the clique to which $s_{t_k}$ and $r_{t_k}$ both belong is discovered, neither $s_{t_k}$ nor $r_{t_k}$ can occur in a future query $\{s_t, r_t\}$) that discovers a new clique. Therefore, in order to have $\mathbb{I}\{D_t \wedge P_t \geq d/2\} = 1$ for $N > \frac{d}{2}$ times, at least

$$\binom{N}{2} \geq \frac{d^2}{8}$$

queries must be made, since each one of the other $N - 1 \geq \frac{d}{2}$ discovered cliques can contribute with at most a query to making $P_t \geq \frac{d}{2}$. So, it takes at least $B \geq \frac{d^2}{8}$ queries to discover the first $\frac{d}{2}$ cliques of size at least two, which contradicts the lemma's assumption that $B \leq \frac{d^2}{16}$. Therefore, $N \leq \frac{d}{2}$.

Using the same logic as before, in order to have $\mathbb{I}\{D_t \wedge P_t \geq d/2\} = 1$ for $N \leq \frac{d}{2}$ times, at least

$$\frac{d}{2} + \left(\frac{d}{2} - 1\right) + \cdots + \left(\frac{d}{2} - N + 1\right)$$

queries must be made. So, it must be

$$B \geq \sum_{k=1}^{N} \left( \frac{d}{2} - (k-1) \right) = (d+1)\frac{N}{2} - \frac{N^2}{2}$$

or, equivalently, $N^2 - (d+1)N + 2B \geq 0$. Solving this quadratic inequality for $N$, and using the hypothesis $N \leq \frac{d}{2}$, we have that $N \leq \frac{(d+1)-\sqrt{(d+1)^2-8B}}{2}$. Using the assumption that $B \leq \frac{d^2}{50}$ we get that $N \leq \sqrt{2B}$.

We now bound the first term of (21) in expectation. The event $D_t$ is equivalent to $s_t, r_t \in T_i$ for some $i \in \neg L_t \cap \neg R_t$, where for any $S \subseteq \{1, \ldots, d\}$ we use $\neg S$ to denote $\{1, \ldots, d\} \setminus S$.

Let $\mathbb{P}_t = \mathbb{P}(\cdot \mid P_t < d/2)$. For $L', R'$ ranging over all subsets of $\{1, \ldots, d\}$ of size strictly less than $\frac{d}{2}$,

$$\mathbb{P}_t(D_t) = \sum_{L',R'} \sum_{i \in \neg L' \cap \neg R'} \mathbb{P}_t\big(s_t \in T_i \wedge r_t \in T_i \mid L_t = L', R_t = R'\big) \mathbb{P}_t(L_t = L' \wedge R_t = R')$$

$$= \sum_{L',R'} \sum_{i \in \neg L' \cap \neg R'} \mathbb{P}_t\big(s_t \in T_i \mid L_t = L'\big) \mathbb{P}_t\big(r_t \in T_i \mid R_t = R'\big) \mathbb{P}_t(L_t = L' \wedge R_t = R')$$

$$\tag{22}$$

$$= \sum_{L',R'} \sum_{i \in \neg L' \cap \neg R'} \frac{1}{|\neg L'|} \frac{1}{|\neg R'|} \mathbb{P}_t(L_t = L' \wedge R_t = R') \tag{23}$$

$$= \sum_{L',R'} \frac{|\neg L' \cap \neg R'|}{|\neg L'||\neg R'|} \mathbb{P}_t(L_t = L' \wedge R_t = R')$$

$$\leq \frac{2}{d} \,. \tag{24}$$

Equality (22) holds because $P_t = \max\{L_t, R_t\} < \frac{d}{2}$ implies that there are at least two remaining cliques to which $s_t$ and $r_t$ could belong, and each node is independently assigned to one of these cliques. Equality (23) holds because, by definition of $L_t$, the clique of $s_t$ is not in $L_t$, and there were no previous queries involving $s_t$ and a node belonging to a clique in $\neg L_t$ (similarly for $r_t$). Finally, (24) holds because $|\neg L'| \geq \frac{d}{2}$, $|\neg R'| \geq \frac{d}{2}$, and $|\neg L' \cap \neg R'| \leq \min\{|\neg L'|, |\neg R'|\}$. Therefore,

$$\sum_{t=1}^{B} \mathbb{P}\big(D_t \wedge P_t < d/2\big) \leq \sum_{t=1}^{B} \mathbb{P}\big(D_t \mid P_t < d/2\big) \leq \frac{2B}{d} \,.$$

Putting everything together,

$$\mathbb{E}\left[ \sum_{t=1}^{B} \mathbb{I}\{D_t\} \right] \leq \frac{2B}{d} + \sqrt{2B} \,. \tag{25}$$

On the other hand, we have

$$\sum_{t=1}^{B} \mathbb{I}\{D_t\} = \sum_{i=1}^{d} \left( \mathbb{I}\left\{|T_i| \geq \frac{n}{2d}\right\} - \mathbb{I}\{E_i\} \right) = d - \sum_{i=1}^{d} \left( \mathbb{I}\left\{|T_i| < \frac{n}{2d}\right\} + \mathbb{I}\{E_i\} \right) \tag{26}$$

Combining (25) and (26), we get that

$$\sum_{i=1}^{d} \mathbb{P}(E_i) \geq d - \sum_{i=1}^{d} \mathbb{P}\big(|T_i| < \tfrac{n}{2d}\big) - \frac{2B}{d} - \sqrt{2B} \,.$$

By Chernoff-Hoeffding bound, $\mathbb{P}\big(|T_i| < \frac{n}{2d}\big) \leq \frac{1}{d^2}$ for each $i = 1, \ldots, d$ when $n \geq 16d\ln d$. Therefore,

$$\sum_{i=1}^{d} \mathbb{P}(E_i) \geq d - \frac{2B+1}{d} - \sqrt{2B} \,.$$

To finish the proof, suppose on the contrary that $\sum_{i=1}^{d} \mathbb{P}(E_i) \leq \frac{d}{2}$. Then from the inequality above, we would get that

$$\frac{d}{2} \geq d - \frac{2B+1}{d} - \sqrt{2B}$$

which implies $B \geq \left(\frac{2-\sqrt{2}}{4}\right)^2 d^2 > \frac{d^2}{50}$, contradicting the assumptions. Therefore, we must have $\sum_{i=1}^{d} \mathbb{P}(E_i) > \frac{d}{2}$ as required. $\hfill\square$

## 4.2 Proof of Theorem 9

Choose a suitably large $n$ and let $V = [n]$. We partition $V$ in two sets $A$ and $B$, where $|A| = \alpha n$ and $|B| = (1-\alpha)n$; we will eventually set $\alpha = 0.9$, but for now we leave it free to have a clearer proof. The set $A$ is itself partitioned into $k = 1/\varepsilon$ subsets $A_1, \ldots, A_k$, each one of equal size $\alpha n/k$ (the subsets are not empty because of the assumption on $\varepsilon$). The labeling $\sigma$ is the distribution defined as follows. For each $i = 1, \ldots, k$, for each pair $u, v \in A_i$, $\sigma(u, v) = +1$; for each $u, v \in B$, $\sigma(u, v) = -1$. Finally, for each $v \in B$ we have a random variable $i_v$ distributed uniformly over $[k]$. Then, $\sigma(u, v) = +1$ for all $u \in A_{i_v}$ and $\sigma(u, v) = -1$ for all $u \in A \setminus A_{i_v}$. Note that the distribution of $i_v$ is independent of the (joint) distributions of the $i_w$'s for all $w \in B \setminus \{v\}$.

Let us start by giving an upper bound on $\mathbb{E}[\text{OPT}]$. To this end consider the (possibly suboptimal) clustering $\mathcal{C} = \{C_i : i \in [k]\}$ where $C_i = A_i \cup \{v \in B : i_v = i\}$. One can check that $\mathcal{C}$ is a partition of $V$. The expected cost $\mathbb{E}[\Delta_{\mathcal{C}}]$ of $\mathcal{C}$ can be bound as follows. First, note the only mistakes are due to pairs $u, v \in B$. However, for any such fixed pair $u, v$, the probability of a mistake (taken over $\sigma$) is $\mathbb{P}(i_u \neq i_v) = 1/k$. Thus,

$$\mathbb{E}[\text{OPT}] \leq \mathbb{E}[\Delta_0] < \frac{|B|^2}{k} = \frac{(1-\alpha)^2 n^2}{k} \tag{27}$$

Let us now turn to the lower bound on the expected cost of the clustering produced by an algorithm. For each $v \in B$ let $Q_v$ be the total number of distinct queries the algorithm makes to pairs $\{u, v\}$ with $u \in A$ and $v \in B$. Let $Q$ be the total number of queries made by the algorithm; obviously, $Q \geq \sum_{v \in B} Q_v$. Now let $S_v$ be the indicator variable of the event that one of the queries involving $v$ returned $+1$. Both $Q_v$ and $S_v$ as random variables are a function of the input distribution and of the choices of the algorithm. The following is key:

$$\mathbb{P}(S_v \wedge Q_v < {}^{k}\!/2) < \frac{1}{2} \tag{28}$$

The validity of (28) is seen by considering the distribution of the input limited to the pairs $\{u, v\}$. Indeed, $S_v \wedge Q_v < {}^{k}\!/2$ implies the algorithm discovered the sole positive pair involving $v$ in less than $k/2$ queries. Since there are $k$ pairs involving $v$, and for any fixed $j$ the probability (taken over the input) that the algorithm finds that particular pair on the $j$-th query is exactly $1/k$. Now,

$$\mathbb{P}(S_v \wedge Q_v < {}^{k}\!/2) + \mathbb{P}(\overline{S_v} \wedge Q_v < {}^{k}\!/2) + \mathbb{P}(Q_v \geq {}^{k}\!/2) = 1 \tag{29}$$

and therefore

$$\mathbb{P}(\overline{S_v} \wedge Q_v < {}^{k}\!/2) + \mathbb{P}(Q_v \geq {}^{k}\!/2) > \frac{1}{2} \tag{30}$$

Let us now consider $R_v$, the number of mistakes involving $v$ made by the algorithm. We analyse $\mathbb{E}[R_v \mid \overline{S_v} \wedge Q_v < {}^{k}\!/2]$. For all $i \in [k]$ let $Q_v^i$ indicate the event that, for some $u \in A_i$, the algorithm queried the pair $\{u, v\}$. Let $I = \{i \in [k] : Q_v^i = 0\}$; thus $I$ contains all $i$ such that the algorithm did not query any pair $u, v$ with $u \in A_i$. Suppose now the event $\overline{S_v} \wedge Q_v < {}^{k}\!/2$ occurs. On the one hand, $\overline{S_v}$ implies that:

$$\mathbb{P}(\sigma(u, v) = +1 \mid I) = \begin{cases} {}^{1}\!/|I| & u \in A_i, i \in I \\ 0 & u \in A_i, i \in [k] \setminus I \end{cases} \tag{31}$$

Informally speaking, this means that the random variable $i_v$ is distributed uniformly over the (random) set $I$. Now observe that, again conditioning on the joint event $\overline{S_v} \wedge Q_v < {}^{k}\!/2$, whatever label $s$ the algorithm assigns to a pair $u, v$ with $u \in A_i$ where $i \in I$, the distribution of $\sigma(u, v)$ is independent

of $s$. This holds since $s$ can obviously be a function only of $I$ and of the queries made so far, all of which returned $-1$, and possibly of the algorithm's random bits. In particular, it follows that:

$$\mathbb{P}(\sigma(u,v) \neq s \mid I) \geq \min\left\{1/|I|, 1 - 1/|I|\right\} \tag{32}$$

However, $Q_v < k/2$ implies that $|I| \geq k - Q_v > k/2 = 2/\varepsilon > 2$, which implies $\min\{1/|I|, 1-1/|I|\} \geq 1/|I|$. Therefore, $\mathbb{P}(\sigma(u,v) \neq s \mid I) \geq 1/|I|$ for all $u \in A_i$ with $i \in I$.

We can now turn to back to $R_v$, the number of total mistakes involving $v$. Clearly, $R_v \geq \sum_{i=1}^{k}\sum_{u \in A_i} \mathbb{I}\{\sigma(u,v) \neq s\}$. Then:

$$\mathbb{E}[R_v \mid E] = \mathbb{E}\left[\sum_{i=1}^{k}\sum_{u \in A_i} \mathbb{I}\{\sigma(u,v) \neq s\} \,\Big|\, \overline{S_v} \wedge Q_v < k/2\right] \tag{33}$$

$$= \mathbb{E}\left[\mathbb{E}\left[\sum_{i=1}^{k}\sum_{u \in A_i} \mathbb{I}\{\sigma(u,v) \neq s\} \,\Big|\, I\right] \Big|\, \overline{S_v} \wedge Q_v < k/2\right] \tag{34}$$

$$\geq \mathbb{E}\left[\mathbb{E}\left[\sum_{i \in I}\sum_{u \in A_i} \mathbb{I}\{\sigma(u,v) \neq s\} \,\Big|\, I\right] \Big|\, \overline{S_v} \wedge Q_v < k/2\right] \tag{35}$$

$$\geq \mathbb{E}\left[\mathbb{E}\left[\sum_{i \in I}\sum_{u \in A_i} \frac{1}{|I|} \,\Big|\, I\right] \Big|\, \overline{S_v} \wedge Q_v < k/2\right] \tag{36}$$

$$= \mathbb{E}\left[\mathbb{E}\left[\frac{\alpha n}{k}\right] \Big|\, \overline{S_v} \wedge Q_v < k/2\right] \tag{37}$$

$$= \frac{\alpha n}{k} \tag{38}$$

And therefore:

$$\mathbb{E}[R_v] \geq \mathbb{E}[R_v \mid \overline{S_v} \wedge Q_v < k/2] \cdot \mathbb{P}(\overline{S_v} \wedge Q_v < k/2)$$
$$> \frac{\alpha n}{k} \cdot \mathbb{P}(\overline{S_v} \wedge Q_v < k/2)$$

This concludes the bound on $\mathbb{E}[R_v]$. Let us turn to $\mathbb{E}[Q_v]$. Just note that:

$$\mathbb{E}[Q_v] \geq \frac{k}{2} \cdot \mathbb{P}(Q_v \geq k/2) \tag{39}$$

By summing over all nodes, we obtain:

$$\mathbb{E}[Q] \geq \sum_{v \in B} \mathbb{E}[Q_v] \geq \frac{k}{2}\left(\sum_{v \in B} \mathbb{P}(Q_v \geq k/2)\right) \tag{40}$$

$$\mathbb{E}[\Delta] \geq \sum_{v \in B} \mathbb{E}[R_v] > \frac{\alpha n}{k}\left(\sum_{v \in B} \mathbb{P}(\overline{S_v} \wedge Q_v < k/2)\right) \tag{41}$$

to which, by virtue of (30), applies the constraint:

$$\left(\sum_{v \in B} \mathbb{P}(Q_v \geq k/2)\right) + \left(\sum_{v \in B} \mathbb{P}(\overline{S_v} \wedge Q_v < k/2)\right) > |B|\frac{1}{2} = \frac{(1-\alpha)n}{2} \tag{42}$$

This constrained system gives the bound. Indeed, by (40), (41) and (42), it follows that if $\mathbb{E}[Q] < \frac{k}{2}\frac{(1-\alpha)n}{4} = \frac{(1-\alpha)nk}{8}$ then $\mathbb{E}[\Delta] > \frac{\alpha n}{k}\frac{(1-\alpha)n}{2} = \frac{\alpha(1-\alpha)n^2}{4k}$. It just remains to set $\alpha$ and $k$ properly so to get the statement of the theorem.

Let $\alpha = 9/10$ and recall that $k = 1/\varepsilon$. Then, first, $\frac{(1-\alpha)nk}{8} = \frac{nk}{80} = \frac{n}{80\,\varepsilon}$. Second, (27) gives $\mathbb{E}[\text{OPT}] < \frac{(1-\alpha)^2 n^2}{k} = \frac{n^2}{100k} = \frac{\varepsilon n^2}{100}$. Third, $\frac{\alpha(1-\alpha)n^2}{4k} = \frac{9n^2}{400k} = \frac{9\varepsilon n^2}{400} > \mathbb{E}[\text{OPT}] + \frac{\varepsilon n^2}{80}$. The above statement hence becomes: if $\mathbb{E}[Q] < \frac{n}{80\varepsilon}$, then $\mathbb{E}[\Delta] > \mathbb{E}[\text{OPT}] + \frac{\varepsilon n^2}{80}$. An application of Yao's minimax principle completes the proof.

As a final note, we observe that for every $c \geq 1$ the bound can be put in the form $\mathbb{E}[\Delta] \geq c \cdot \mathbb{E}[\text{OPT}] + \Omega(n^2 \varepsilon)$ by choosing $\alpha \geq c/(c + 1/4)$.

# 5 Supplementary Material for Section 7

We report the complete experimental evaluation of ACC including error bars (see the main paper for a full description of the experimental setting). The details of the datasets are found in Table 1.

Table 1: Description of the datasets.

| Datasets | Type | $|V|$ | #Clusters |
|---|---|---|---|
| captchas | Real | 244 | 69 |
| cora | Real-world | 1879 | 191 |
| gym | Real | 94 | 12 |
| landmarks | Real | 266 | 12 |
| skew | Synthetic | 900 | 30 |
| sqrt | Synthetic | 900 | 30 |

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

Figure 1: Clustering cost vs. number of queries.

(a) skew.

(b) sqrt.

(c) cora.

(d) landmarks.

(e) gym.

(f) captchas.