[Reviews · NeurIPS 2019]

Reviewer 1



This paper studies correlation clustering in an active learning setting where the learner does not query for all the {n choose 2} pairs of vertices. In the standard setting, the algorithm KwikCluster of Charikar et al. achieves a clustering cost of 3OPT where OPT is the cost of the best clustering. Here the cost of the clustering is defined as the number of pairs labeled +1 and put in different clusters plus the number of pairs labeled -1 and put in the same cluster. The main contribution of the paper is a variant ACC of KwikCluster that achieves a cost of 3OPT + O(n^3/Q) where Q is an upper bound on the number of queries made by the algorithm. The authors also prove a matching lower bound on the error of any randomized algorithm that makes Q queries. Next the paper shows that the ACC algorithms also comes with per cluster recovery guarantee when the clusters in the optimal solution are near cliques. This is formalized as the tight-knit condition. Finally, the authors model the correlation clustering problem as an agnostic learning problem over pairs of vertices and show that ERM will achieve a query bound of OPT + n^2eps using n/eps^2 queries. In section 7 the authors show experimental results comparing the performance of ACC as the query budget increases for different values of OPT. Techniques: --------------- In order to get the active variant, the strategy is to run KwikCluster where at each step a node x is chosen at random. However, as opposed to KwikCluster, only a subset of random chosen points are compared to x. If any of those queries return 1, then x is compared to every data point, otherwise a singleton cluster containing x is output. The number of points to compare at step r is defined by a function f(r), for instance one could f(r) to be r^.2. The proof decomposes the number of mistakes into ones that form a "bad" triangle and can then be bounded as in Charikar et al. The others where (u,v) go together but u or v is output as singletons are unlikely is u belongs to a large cluster. In my opinion the ACC algorithm and the corresponding lower bound form the key contribution of the paper. The proof technique extends the argument of Charikar et al. in a clever way and the proposed algorithm could be practically useful. The connection to agnostic learning is fairly standard and the cluster recovery statements make fairly strong assumptions on the graph. In general the paper is well written.

Reviewer 2



In this paper, authors have tried to solve the ccorrelation clustering problem in a setting where the (binary) pairwise similarities are not known apriori. Authors consider an active learning model where the pairwise similarity value is queried from an oracle. The authors refer to a standard algorithm namely KwikCLuster and modify it to come up with a sampling based mechanism. The ACC algorithm proposed by the authors is same Kwikcluster in the worst case and authors prove some cost bounds of their techniques. [W1] I have some concerns about the motivation of this work: The authors assume that the similarity function has binary output. This means that their work is highly focussed towards entity resolution based setting where the ground truth consists of a collection of entities and the goal is to resolve the entities. If this is the case, the authors should perform a thorough job of comparing with most recent works on entity resolution. I think even though [20] is not suited for adversarial noise, it can still work in this setting and might perform better. Especially in experiments because authors use independent probability p to flip the edges. Calling the queries 'similarity queries' seems to give a wrong impression as it is binary Yes/No query. [W2] Theorem 1 does not seem tight eg. when Q=n^2, we get expected error is less than 3OPT+O(n), whereas we know that the expected error should be 3OPT as it is same as Kwikcluster. [W3] Experiments: - The data sets are really small with no real crowd experiments validating if similarity queries are easy to answer for a user or not. The authors have used Yes/No answers from entity resolution crowdsourcing experiments as 1/0 similarity values and it does not seem like a fair justification of the proposed querying oracle. - There is no comparison with any baseline (Not even kwikcluster). I think some of the robust techniques in data base comunity to reduce the number of queries in such setting might be interesting baselines to help appreciate ACC. - In the plots, authors dont mention OPT (or 3OPT if calculating opt was not possible). - It seems like ACC takes close to 60K queries to identify the optimal clusters for cora data set. I think experiments need a lot of work. [W4] Authors should analyze the performance if the edge labels (binary similarities) are flipped with some probability. Overall, I think the work has some promise but it needs a lot of work.

Reviewer 3



The authors study correlation clustering where the similarity scores are hidden and can only be accessed using a query to an oracle. The goal is to find a good partition of the graph with small number of queries. The exposition is relatively clean and done in a careful manner. Regarding the results however in most of the results the queries required for the guarantees of the theorem are prohibitively large close to n or n^2 in most cases. I believe this is a crucial problem with the model considered here. I think the model of Approximate Correlation Clustering with using Same Cluster queries by Nir Ailon, Anup Bhattacharya, Ragesh Jaiswal and Approximate Clustering with Same Cluster queries by Nir Ailon, Anup Bhattacharya, Ragesh Jaiswal, Amit Kumar is much more reasonable and there they get guarantees compared to k the number of clusters in the optimum solution that is usually much fewer than n. Given these papers, I think the originality of the query model is decreased and the significance of the results as well. Furthermore, related work is not done in a detailed manner (see below in the Improvements section). Typos/Comments to improve exposition: Abstract: When Q = n^2 that the suggested algorithm matches the guarantees of KwikCluster but isn't there an additional O(n) term? What is a quick explanation for that?

[Author Response · NeurIPS 2019]

We thank the reviewers for their work. We ask Reviewer 2 and Reviewer 3 to reconsider their decision in the light of our feedback.

## Reviewer 1

**1. The assumptions used in cluster recovery are fairly strong in my opinion. It is not clear to what extent they are required.** The assumptions are necessary, at least in part. For $\varepsilon \geq \frac{1}{2}$, changing the sign of $\varepsilon C$ edges of a node in a cluster $C$ makes $C$ not optimal anymore, so no cluster recovery algorithm should output it. Moreover, this holds for exact recovery, while for recovery *in expectation* our assumptions are weaker.

## Reviewer 2

**[W1] [...] The authors assume that the similarity function has binary output. This means that their work is highly focussed towards entity resolution based setting where the ground truth consists of a collection of entities and the goal is to resolve the entities. If this is the case, the authors should perform a thorough job of comparing with most recent works on entity resolution. [...]**
**[W3] The data sets are really small with no real crowd experiments validating if similarity queries are easy to answer for a user or not. The authors have used Yes/No answers from entity resolution crowdsourcing experiments as 1/0 similarity values and it does not seem like a fair justification of the proposed querying oracle.** Our paper is not focussed on entity resolution and does not propose a query oracle. It is a theoretical study of correlation clustering under the lenses of active learning theory, with a standard query model that has been used for decades. This model uses a binary similarity function exactly because this captures entity resolution (ER) and many other problems, as we say clearly in the introduction. Thus, we don't think our approach is "focused on" ER. It is the CC framework which *comes from* ER-type problems. Because CC has been studied for over 15 years, we do not feel compelled to provide any further justification. Similarly, we do not see any reason to compare against ER: our paper focuses on the fundamental algorithmic mechanisms without referring to any specific applications domain,

**[W2] Thm 1 does not seem tight e.g. when $Q = n^2$, we get expected error less than $3\mathrm{OPT} + O(n)$, whereas we know that the expected error should be $3\mathrm{OPT}$ as it is same as KwikCluster.** We do not understand the reviewer's comment. The last three lines of the theorem state that, in the special case $Q = \binom{n}{2}$, our algorithm *exactly* achieves error $3\mathrm{OPT}$ like KwikCluster does, with no additive error term. We will clarify this should the paper be accepted.

**[W3] There is no comparison with any baseline (Not even kwikcluster).** We do not understand the reviewer's comment. The experiments report *precisely* a comparison with KwikCluster (caption of Figure 1, last line).

**[W3] In the plots, authors do not mention $\mathrm{OPT}$ (or $3\mathrm{OPT}$ if calculating $\mathrm{OPT}$ was not possible).** We do not know $\mathrm{OPT}$ since computing it is NP-hard, and the same obviously holds for $3\mathrm{OPT}$. However, the plots show the cost achieved by KwikCluster, which in expectation is precisely between $\mathrm{OPT}$ and $3\mathrm{OPT}$.

**[W4] Authors should analyze the performance if the edge labels (binary similarities) are flipped with some probability.** We are unsure about what the reviewer means. If one takes an instance and flips the signs of some edges, then one simply obtains another instance, whose optimum will be in general different, and to which our results still apply. We provide cluster recovery results even under adversarial edge flips, if this is what the reviewer meant.

## Reviewer 3

**[...] the queries [...] are prohibitively large close to $n$ or $n^2$ in most cases. I believe this is a crucial problem with the model considered here. The model of Ailon [...] is much more reasonable and there they get guarantees compared to $k$ the number of clusters in the optimum solution that is usually much fewer than $n$. [...] the originality of the query model is decreased and the significance of the results as well.** We strongly disagree with this comment. Counting the number of similarity queries is the standard model for the analysis of query complexity, sublinear algorithms, and active learning on graphs, and we do not claim any originality on it. It captures the inherent effort of obtaining similarities, as this is a major bottleneck in practice (as Reviewer 2 points out, too). Ailon's model is about "computational complexity with advice": it gives all the $\binom{n}{2}$ similarities for free, and then equips the algorithm with an oracle so powerful to turn an APX-hard problem into a polynomial one! Only the queries to this oracle are counted in the analysis. We do not find this "much more reasonable" than our model; it is simply different. And, to be fair, Alon's algorithm needs *more* queries than us: $\binom{n}{2}$ similarity queries plus $O(k^{14}/\varepsilon^6)$ oracle queries, even fixing $k$. We will include this discussion in the related work section should the paper be accepted.

**When $Q = n^2$ that the suggested algorithm matches the guarantees of KwikCluster but is not there an additional $O(n)$ term?** Please see [W2] of Reviewer 2.

[Meta-Review · NeurIPS 2019]

There was considerable discussion about this paper in the post rebuttal period. The reviewers agree that the author rebuttal addresses/clarifies their concerns -- particularly the novelty of the paper in the setting considered and the fact that this is primarily a theoretical paper. Nevertheless the authors would be well-advised to add the clarifications they mentioned in the rebuttal in greater detail in the camera-ready version.